# ONE-SHOT GENERATIVE DOMAIN ADAPTATION

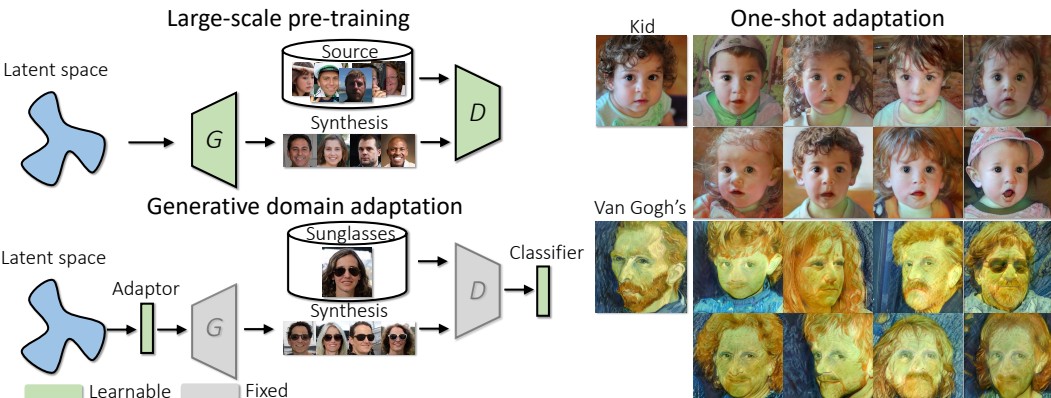

Figure 1: **Diagram of one-shot generative domain adaptation. Left:** The overall framework, where a GAN model pre-trained on the large-scale source data is transferred to the target domain with *only one training sample*. A lightweight attribute adaptor and attribute classifier are introduced to the frozen generator and discriminator respectively. **Right:** Realistic and *highly diverse* synthesis results after adapting the pre-trained model with different targets. Here, our efficient design manages to transfer both semantic attributes and artistic styles *within a few minutes*.

## ABSTRACT

This work aims at transferring a Generative Adversarial Network (GAN) pre-trained on one image domain to a new domain *referring to as few as just one target image*. The main challenge is that, under limited supervision, it is extremely difficult to synthesize photo-realistic and highly diverse images, while acquiring representative characters of the target. Different from existing approaches that adopt the vanilla fine-tuning strategy, we import two lightweight modules to the generator and the discriminator respectively. Concretely, we introduce an *attribute adaptor* into the generator yet freeze its original parameters, through which it can reuse the prior knowledge to the most extent and hence maintain the synthesis quality and diversity. We then equip the well-learned discriminator backbone with an *attribute classifier* to ensure that the generator captures the appropriate characters from the reference. Furthermore, considering the poor diversity of the training data (*i.e.*, as few as only one image), we propose to also constrain the diversity of the generative domain in the training process, alleviating the optimization difficulty. Our approach brings appealing results under various settings, *substantially* surpassing state-of-the-art alternatives, especially in terms of synthesis diversity. Noticeably, our method works well even with large domain gaps, and robustly converges *within a few minutes* for each experiment.

## 1 INTRODUCTION

Generative Adversarial Network (GAN) (Goodfellow et al., 2014), consisting of a generator and a discriminator, has significantly advanced image synthesis yet relies on a large number of training samples (Karras et al., 2018; 2019; 2020b; Brock et al., 2018). Many attempts have been made to train GANs from scratch with limited data (Zhang & Khoreva, 2019; Zhao et al., 2020d;b; Karras et al., 2020a; Yang et al., 2021a), but it still requires hundreds or thousands of images to get a

satisfying performance. Sometimes, however, we may have as few as only one single image as the reference, like the masterpiece Mona Lisa from Leonardo da Vinci. Under such a case, learning a generative model with both good quality and high diversity becomes extremely challenging.

Domain adaptation is a commonly used technique that can apply an algorithm developed on one data domain to another (Csurka, 2017). Prior works (Wang et al., 2018; Noguchi & Harada, 2019; Wang et al., 2020; Mo et al., 2020; Zhao et al., 2020a; Li et al., 2020; Robb et al., 2020) have introduced this technique to GAN training to alleviate the tough requirement on the data scale. Typically, they first train a large-scale model in the source domain with adequate data, and then transfer it to the target domain with only a few samples. A common practice is to fine-tune both the generator and the discriminator on the target dataset until the generator produces samples conforming to the target domain. To stabilize the fine-tuning process and improve the generation quality and diversity, existing approaches propose to tune partial parameters (Noguchi & Harada, 2019; Mo et al., 2020; Robb et al., 2020) and introduce some regularizers (Li et al., 2020; Ojha et al., 2021), but the overall adaptation strategy remains. When there is only one image from the target domain, these methods would fell short of synthesis diversity, producing very similar images.

Recall that the pre-trained model can already produce highly diverse images from the source domain. It makes us wonder what indeed causes the diversity drop in the adaptation process. We argue that directly tuning the model weights will result in the loss of the prior knowledge gained from the large-scale data. On the other hand, however, when adapting the model to the target domain, most variation factors (*e.g.*, gender and pose of human faces) could be reused. These observations help raise a question: is it possible to simply focus on the most representative characters of the reference image while inheriting all the other knowledge from the source model?

In this work, we develop a method, called **GenDA**, for one-shot Generative Domain Adaptation. In particular, we design a lightweight module connecting the latent space and the synthesis network. We call this module an *attribute adaptor* since it helps adapt the generator with the attributes of the target image. Unlike the conventional fine-tuning strategy, we freeze the parameters of the original generator and merely optimize the attribute adaptor during training. Thereby, we manage to reuse the prior knowledge learned by the source model and hence inherit the synthesis quality and, more importantly, diversity. Meanwhile, we employ the discriminator to compete with the generator via a domain-specific attribute classification. In this way, the generator is forced to capture the most representative attributes from the reference, or otherwise, the discriminator would spot the discrepancy. However, instead of directly tuning the original discriminator, we freeze its entire backbone and introduce a lightweight *attribute classifier* on top of that. Similar to the generator, the discriminator has also learned rich knowledge in its pre-training. Since the synthesized images before and after adaptation share most visual concepts (*e.g.*, a face model would still produce faces after domain transfer), the discriminator can be reused as a well-learned feature extractor. Therefore, we simply train the attribute classifier to help guide the generator. Furthermore, since there is only one training sample (which means no diversity in the target domain), we propose to also constrain the diversity of the generative domain by truncating the latent distribution during training. Intuitively, learning a one-to-one mapping would be easier than learning a many-to-one mapping. Such a design mitigates the optimization difficulty and further improves the synthesis quality.

We evaluate our approach through extensive experiments on human faces and outdoor churches. Given only one training image, GenDA can convincingly adapt the source model to the target domain with sufficiently high quality and diversity. Such an adaptation includes both attribute-level and style-level, as shown in Fig. 1. Our method outperforms the state-of-the-art competitors *by a large margin* both qualitatively and quantitatively. We also show that, when the number of the samples available in the target domain increases, GenDA can filter out the individual attributes and only preserve their common characters (see Fig. 4). Noticeably, GenDA works well for the extreme cases where there is a large domain gap, like transferring the characters of Mona Lisa to churches (see Fig. 5). Besides, thanks to the lightweight design of both the attribute adaptor and the attribute classifier, GenDA can finish each adaptation experiment *within a few minutes*.

## 2 METHODOLOGY

The primary goal of this work is to transfer a pre-trained GAN to synthesize images conforming to a new domain with as few as only one reference image. Due to the limited supervision, it is

challenging to ensure both high quality and large diversity of the synthesis. Intuitively, according to the rationale of GANs (*i.e.*, adversarial training between the generator and the discriminator), the discriminator can easily memorize that only the reference image is real while all others are fake. In this way, to fool the discriminator, the generator may have to learn to produce images highly alike the reference, resulting in a poor synthesis diversity. To mitigate this problem, we propose a new adaptation algorithm, different from the previous fine-tuning scheme. Concretely, we (i) interpose an *attribute adaptor* between the latent space and the generator to help acquire the most representative characters from the target image; (ii) augment the discriminator backbone with an *attribute classifier* to guide the generator to make appropriate adjustments; and (iii) propose a *diversity-constraint* training strategy. Before going into technical details, we first give some preliminaries of GANs.

## 2.1 PRELIMINARIES

Generative Adversarial Network (GAN) (Goodfellow et al., 2014) is formulated as a two-player game between a generator and a discriminator. Given a collection of observed data $\{\mathbf{x}_i\}_{i=1}^{N}$ with $N$ samples, the generator $G(\cdot)$ aims at reproducing the real distribution $\mathcal{X}$ via randomly sampling latent codes $\mathbf{z}$ subject to a pre-defined latent distribution $\mathcal{Z}$. As for the discriminator $D(\cdot)$, it targets at differentiating the real data $\mathbf{x}$ and the synthesized data $G(\mathbf{z})$ as a bi-classification task. These two models are jointly optimized by competing with each other, as

$$\mathcal{L}_G = -\mathbb{E}_{\mathbf{z} \in \mathcal{Z}}[\log(D(G(\mathbf{z})))], \tag{1}$$

$$\mathcal{L}_D = -\mathbb{E}_{\mathbf{x} \in \mathcal{X}}[\log(D(\mathbf{x}))] - \mathbb{E}_{\mathbf{z} \in \mathcal{Z}}[\log(1 - D(G(\mathbf{z})))]. \tag{2}$$

After the training converges, the generator is expected to produce images as realistic as the training set, so that the discriminator cannot distinguish them anymore.

In this work, we start with a GAN model that is well trained on a source domain $\mathcal{X}^{src}$, and work on adapting it to a target domain $\mathcal{X}^{dst} = \{\mathbf{x}^{dst}\}$ that has only one image. In fact, it is ambiguous to define a "domain" using one image. We hence expect the model to acquire the most representative characters from the reference image. Taking face synthesis as an example, the character can include facial attributes (*e.g.*, age or wearing sunglasses) and artistic styles, as shown in Fig. 1.

## 2.2 ONE-SHOT GENERATIVE DOMAIN ADAPTATION

A common practice to transfer GANs is to simultaneously tune the generator and the discriminator on the target dataset (Wang et al., 2018). However, as discussed above, the transferring difficulty increases drastically when given only one training sample. Existing methods attempt to address this issue by reducing the number of learnable parameters (Mo et al., 2020; Robb et al., 2020) and introducing training regularizers (Ojha et al., 2021). Even so, the overall fine-tuning scheme (*i.e.*, directly tuning $G(\cdot)$ and $D(\cdot)$) remains and the diversity is low. Differently, we propose a new adaptation pipeline to preserve the synthesis diversity, which includes an attribute adaptor, an attribute classifier, and a diversity-constraint training strategy. Details are introduced as follows.

**Attribute Adaptor.** Prior works have found that, a well-learned generator is able to encode rich semantics to produce diverse images (Shen et al., 2020; Yang et al., 2021b). For instance, a face synthesis model could capture the variation factors like gender, age, wearing glasses, lighting, *etc*. Ideally, this knowledge should be sufficiently reused as much as possible in the target domain, and then the synthesis diversity can be preserved accordingly. In this way, the generator focuses on transferring the most distinguishable characters of the only reference, instead of learning the common variation factors repeatedly and leading to overfitting due to lack of samples. This helps improve the data efficiency significantly, which is vital to the one-shot setting.

According to Xu et al. (2021), the latent code $\mathbf{z}$ can be viewed as the generative feature of $G(\mathbf{z})$ that determines the multi-level attributes of the output image. Motivated by this, we propose to adapt such features regarding the reference image, yet keep the convolutional kernels untouched. Concretely, before feeding the latent code $\mathbf{z}$ to the generator $G(\cdot)$. we propose to first transform it through *a lightweight attribute adaptor* $A(\cdot)$, as

$$\mathbf{z}' = A(\mathbf{z}) = \mathbf{a} \odot \mathbf{z} + \mathbf{b}, \tag{3}$$

where $\odot$ stands for the element-wise multiplication, while $\mathbf{a}$ and $\mathbf{b}$ are the learnable weight and bias respectively. With such a design, the transformed latent code $\mathbf{z}'$ is assumed to carry the sufficient information of the reference, and therefore $G(\mathbf{z}')$ would conform to the target domain $\mathcal{X}^{dst}$.

**Attribute Classifier.** Only having the attribute adaptor cannot guarantee the generator to acquire the representative characters from the training sample. Following the formulation of GANs, we incorporate the discriminator $D(\cdot)$, which is also pre-trained on the source domain, to compete with the generator. In particular, we reuse the backbone $d(\cdot)$ but remove the last real/fake classification head, and then equip it with *a lightweight attribute classifier* $\phi(\cdot)$. Given an image $\mathbf{x}$, either the reference image $\mathbf{x}^{dst}$ or a transferred synthesis $G(A(\mathbf{z}))$, the classifier outputs a probability of how likely it possesses the target attribute, as

$$p = \phi(d(\mathbf{x})). \tag{4}$$

However, due to the limited supervision provided by one image, the discriminator can easily memorize the real data, which leads to the overfitting of discriminator as well as the collapse of the generator (Karras et al., 2020a). As discussed above, the generated images before and after domain adaptation are expected to share most variation factors (*i.e.*, a face model remains to produce faces after adaptation). From this viewpoint, the knowledge learned by the discriminator in its pre-training could be also reused. Therefore, unlike existing approaches that fine-tune all or partial parameters of $D(\cdot)$ (Mo et al., 2020; Wang et al., 2020; Li et al., 2020; Ojha et al., 2021), we freeze all parameters of $d(\cdot)$ in the entire training process and merely optimizes $\phi(\cdot)$ to guide $A(\cdot)$ with adequate adjustments. Regarding the one-shot target domain, the mechanism behind the classifier is very similar to Exemplar SVM (Malisiewicz et al., 2011), which also suggests that it is sufficient to obtain a good decision boundary with one positive and many negative samples. Differently, our attribute classifier is learned in an adversarial manner through competing with the attribute adaptor.

**Diversity-constraint Strategy.** Recall that this work targets at generative domain adaptation with only one reference image, which means no diversity of real data. On the contrary, however, the latent code can be sampled randomly and the pre-trained generator can produce highly diverse images from the source domain. From this perspective, it might be challenging to match these two distributions with such a huge diversity gap. To alleviate the optimization difficulty, we propose a diversity-constraint strategy, which retains the diversity of the generator during training. Specifically, we truncate the latent distribution with a strength factor $\beta$, as

$$\mathbf{z}' = A(\beta\mathbf{z} + (1 - \beta)\bar{\mathbf{z}}), \tag{5}$$

where $\bar{\mathbf{z}}$ indicates the mean code. Note that, truncation is a common trick used in the inference of state-of-the-art GANs, like StyleGAN (Karras et al., 2019) and BigGAN (Brock et al., 2018), to improve synthesis quality. Nevertheless, to our best knowledge, this is the first time that truncation is introduced in the training process to preserve the synthesis diversity.

**Full Objective Function.** In summary, the adaptor $A(\cdot)$ and the classifier $\phi(\cdot)$ are trained with

$$\mathcal{L}_A = -\mathbb{E}_{\mathbf{z}\in\mathcal{Z}}[\log\left(\phi(d(G(\mathbf{z}')))\right)], \tag{6}$$

$$\mathcal{L}_\phi = -\mathbb{E}_{\mathbf{x}\in\mathcal{X}^{src}}[\log\left(\phi(d(\mathbf{x}))\right)] - \mathbb{E}_{\mathbf{z}\in\mathcal{Z}}[\log\left(1 - \phi(d(G(\mathbf{z}')))\right)]. \tag{7}$$

## 3 Experiments

We evaluate the proposed method on multiple datasets and settings. In Sec. 3.1, we focus on one-shot generative domain adaptation. Quantitative and qualitative results indicate that the proposed GenDA can produce much more diverse and photo-realistic images than previous alternatives. Interestingly, the shared representative attributes of multiple shots could be also captured and transferred from the source to the target domain. Sec. 3.2 presents the quantitative comparison to prior approaches, demonstrating the effectiveness of our GenDA under the general few-shot setting. Additionally, when there exists a large domain gap, GenDA can still synthesize reasonable outputs in Sec. 3.3. Noticeably, the comprehensive ablation studies of each component are available in **Appendix**.

**Implementation.** We choose the state-of-the-art StyleGAN2 (Karras et al., 2020b) as our base GAN model, following Ojha et al. (2021). StyleGAN2 proposes a more disentangled $\mathcal{W}$ space in addition to the native latent space $\mathcal{Z}$. Here, our attribute adaptor is deployed on the $\mathcal{W}$ space. $\bar{\mathbf{z}}$ in Eq. (5) becomes $\mathbf{w}_{avg}$, which is a statistical average in the training of the source generator. To alleviate the overfitting problem, we apply differentiable augmentation on both the reference image and the adapted synthesis (Zhao et al., 2020b; Karras et al., 2020a). In particular, we adopt the augmentation pipeline from StyleGAN2-ADA (Karras et al., 2020a) and linearly increase the

Table 1: **Quantitative comparison on one-shot adaptation** between FreezeD (Mo et al., 2020), Cross-Domain (Ojha et al., 2021), and our proposed GenDA. Evaluation metrics include FID (lower is better), precision (higher means better quality), and recall (higher means higher diversity). All results are averaged over 5 training shots.

| Methods | FID | Prec. | Recall | FID | Prec. | Recall | FID | Prec. | Recall |
|---|---|---|---|---|---|---|---|---|---|
| FreezeD | 147.91 | 0.61 | 0.012 | 87.32 | 0.93 | 0.000 | 102.11 | 0.52 | 0.009 |
| Cross-Domain | 146.74 | 0.53 | 0.000 | 91.92 | 0.80 | 0.000 | 90.42 | 0.19 | 0.000 |
| **GenDA (ours)** | 80.16 | 0.74 | 0.033 | 44.96 | 0.76 | 0.067 | 87.55 | 0.16 | 0.053 |
| | (a) Babies | | | (b) Sunglasses | | | (c) Sketches | | |

Figure 2: **Qualitative comparison on one-shot adaptation** between FreezeD (Mo et al., 2020), Cross-Domain (Ojha et al., 2021), and our proposed GenDA. The first column shows the reference images. GenDA *significantly* outperforms the other competitors from the diversity perspective.

augmentation strength from 0 to 0.6. We use Adam optimizer (Kingma & Ba, 2014) for training and the learning rates are set as $1.25e^{-4}$ and $2.5e^{-4}$ for $A(\cdot)$ and $\phi(\cdot)$, respectively. All experiments are stopped when the discriminator has seen $200K$ real samples. The last "ToRGB" layer (with $1 \times 1$ convolution kernels) of the generator is tuned together with the adaptor $A(\cdot)$ to make sure aligning the synthesis color space to the reference image. The hyper-parameter $\beta$ in Eq. (5) is set as 0.7.

**Datasets.** Our source models are pre-trained on large-scale datasets (*e.g.*, FFHQ (Karras et al., 2019) and LSUN Church (Yu et al., 2015)) with resolution $256 \times 256$. We collect target images from FFHQ-babies, FFHQ-sunglasses, face sketches (Ojha et al. (2021)), and samples from masterpieces (*i.e.*, Mona Lisa and Van Gogh's houses) and Artistic-Faces dataset.

**Evaluation Metrics.** Fréchet Inception Distance (FID) (Heusel et al., 2017) serves as the main metric. Akin to Ojha et al. (2021), we always calculate the FID between 5,000 generated images and all the validation sets. Additionally, we report the precision and recall metric (Kynkäänniemi et al., 2019) to measure the quality and diversity respectively.

## 3.1 ONE-SHOT DOMAIN ADAPTATION

**Comparison with Existing Alternatives.** For one-shot generative domain adaptation, we compare against FreezeD (Mo et al., 2020) and Cross-Domain (Ojha et al., 2021). Noticeably, the implemented FreezeD is further improved by the data augmentations, which is already a challenging baseline. Considering the variance brought by different single shots, we calculate all metrics over 5 training shots. As shown in Tab. 1, our GenDA remains to surpass multiple approaches by a clear gap from the perspective of FID. Besides, to further compare different methods from the views of image quality and diversity, we also report the precision and recall. Intuitively, a higher precision means

Target                  Source domain: FFHQ Face

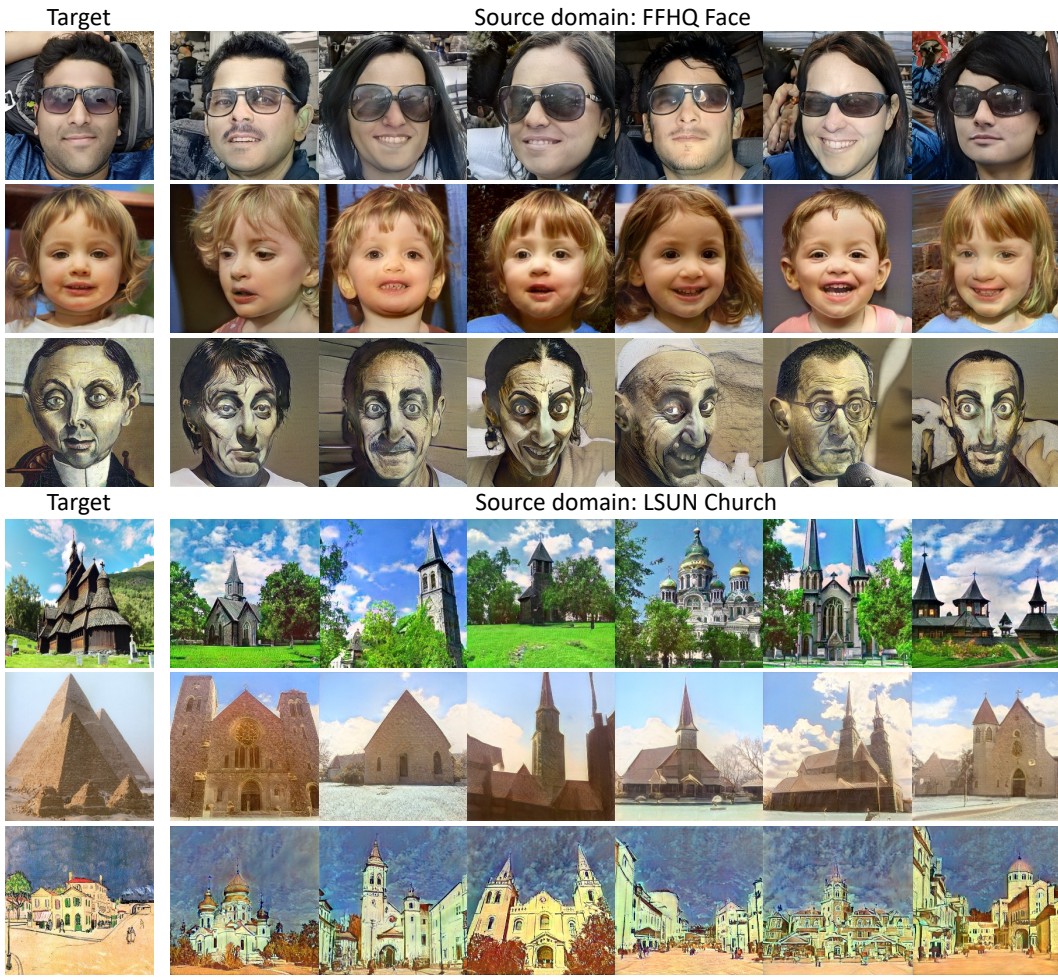

Target                  Source domain: LSUN Church

Figure 3: **One-shot adaptation with different target samples.** GenDA manages to capture the representative characters from the given reference, such as sunglasses, artistic style for faces, and vegetation, pyramid material for churches.

higher image quality (more closer to the real sample), while a higher recall indicates higher diversity. Although FreezeD (Mo et al., 2020) and Cross-Domain (Ojha et al., 2021) plausibly achieves better synthesis quality on sunglasses and sketches, their low recalls strongly imply overfitting. Fig. 2 confirms that there is insufficient diversity for such two methods. As a contrast, our GenDA results in the competitive quality and standing-out diversity from the quantitative and qualitative perspectives. In particular, GenDA manages to synthesize a man with a cap which is only available in prior knowledge. Fig. 3 suggests that our GenDA works on transferring both attributes (*sunglasses, gender, vegetation and material*) and artistic styles for face and church model respectively.

**Common Attributes from Multiple References.** There might be a number of representative variation factors in a given face image (*e.g.*, age, gender, smile and sunglasses). Therefore, when transferring a pre-trained generative model on one image, multiple factors could be adapted together. Fig. 4 provides an example of our GenDA on sunglasses. Specifically, we train GenDA on multiple target domains which might contain a single individual with different identities or a pair of images. Compared with the source output (the first row), representative attributes besides sunglasses are also transferred. For instance, the second and third rows suggest that although all individuals wear sunglasses, the target's gender is also adapted to the new synthesis. The third shot could even affect the gender and hairstyles. This reveals that our GenDA could capture multiple representative attributes of the target domain. When the target domain contains more than one shot like the last two rows of Fig. 4, the representative attributes become the common attributes of all individuals, leading to the corresponding results (*i.e.*, sunglasses and gender). Namely, our GenDA is able to capture and adapt the representative attributes no matter how many images the target domain has.

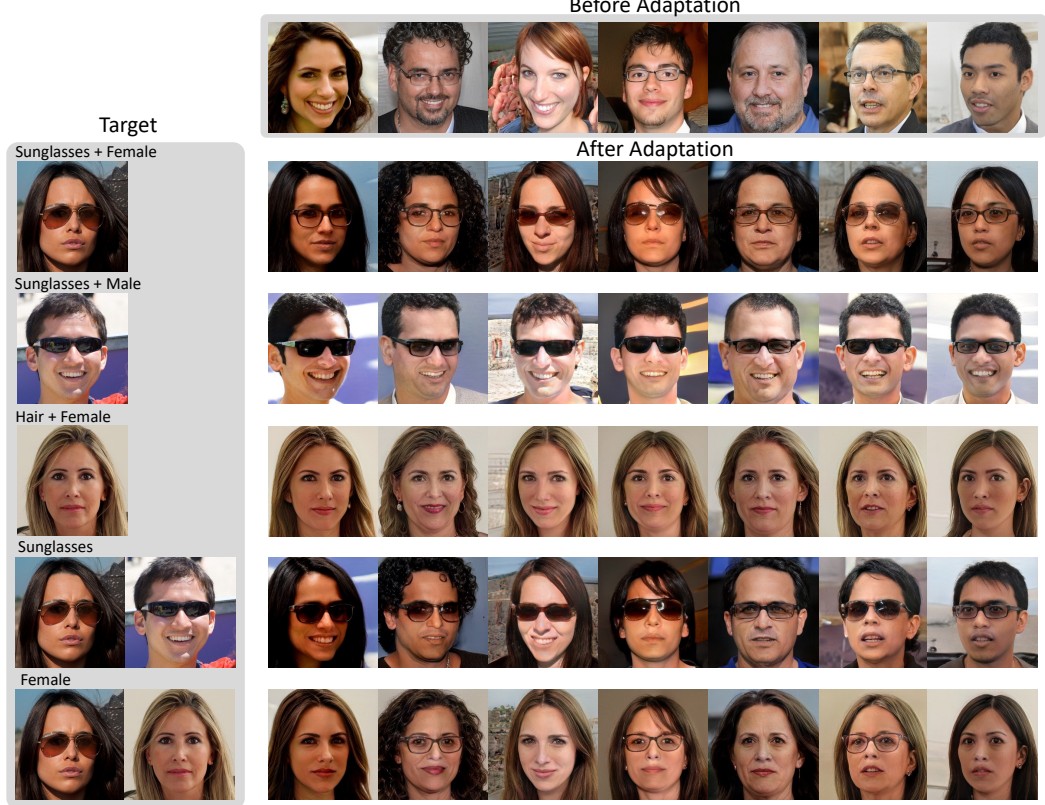

Figure 4: **Transferring the "common" semantic of more than one reference image**. On the left are the training samples, while on the top are the samples synthesized before adaptation. In the remaining rows, images from the same column are produced with the same latent code. Under the settings of two-shot adaptation (*i.e.*, the last two rows), we can tell that (1) in the second last row, all people are wearing eyeglasses (common attribute of the two references) but they have the same gender (divergent attribute) as the original synthesis in the top row. (2) similarly, in the bottom row, all people are female yet the "eyeglasses" attribute is preserved from the original synthesis.

## 3.2 10-shot Domain Adaptation

As our GenDA could capture the representative attributes precisely with an increasing number of target shots, we also compare again prior approaches under the general few-shot adaptation setting. Following the setting of Ojha et al. (2021), we choose 10-shot target domain for comparison.

**Quantitative Comparison.** Multiple baselines are introduced to conduct the quantitative comparisons, including Transferring GANs (TGAN) (Wang et al., 2018), Batch Statistics Adaptation (BSA) (Noguchi & Harada, 2019), FreezeD (Mo et al., 2020), MineGAN (Wang et al., 2020), EWC (Li et al., 2020) and Cross-Domain (Ojha et al., 2021). All these methods adapt a pre-trained source model to a target domain (*e.g.*, babies, sunglasses and sketches). Akin to Cross-domain (Ojha et al., 2021), FID serves as the metric for evaluation.

Tab. 2 presents the 10-shot quantitative comparison on several target domains. Although Cross-Domain (Ojha et al., 2021) has already obtained the competitive results, our method could still significantly improve the performance by a clear margin, resulting in the new state-of-the-art synthesis quality on few-shot adaptation. Specifically, for domains that differ in semantic attributes (*i.e.*, babies and sunglasses), huge gains are obtained by GenDA. It is also worth noting that when the domain gap increases (*i.e.*, from realistic facial images to sketches), the performances of Cross-domain (Ojha et al., 2021) and FreezeD (Mo et al., 2020) become almost identical (45.67 *v.s* 46.54) while substantial improvements of synthesis could remain observed by our method (31.97). It demonstrates the effectiveness of our method on the general few-shot settings.

Table 2: **Quantitative comparison on 10-shot adaptation.** FID (lower is better) serves as the evaluation metric. Numbers in **blue** indicate our improvements over the state-of-the-art alternative (Ojha et al., 2021).

| 10-shot transfer | **Babies** | **Sunglasses** | **Sketches** |
|---|---|---|---|
| TGAN (Wang et al., 2018) | 104.79 | 55.61 | 53.41 |
| TGAN+ADA (Karras et al., 2020a) | 102.58 | 53.64 | 66.99 |
| BSA (Noguchi & Harada, 2019) | 140.34 | 76.12 | 69.32 |
| FreezeD (Mo et al., 2020) | 110.92 | 51.29 | 46.54 |
| MineGAN (Wang et al., 2020) | 98.23 | 68.91 | 64.34 |
| EWC (Li et al., 2020) | 87.41 | 59.73 | 71.25 |
| Cross-Domain (Ojha et al., 2021) | 74.39 | 42.13 | 45.67 |
| **GenDA (ours)** | **47.05** (−27.34) | **22.62** (−19.51) | **31.97** (−13.70) |

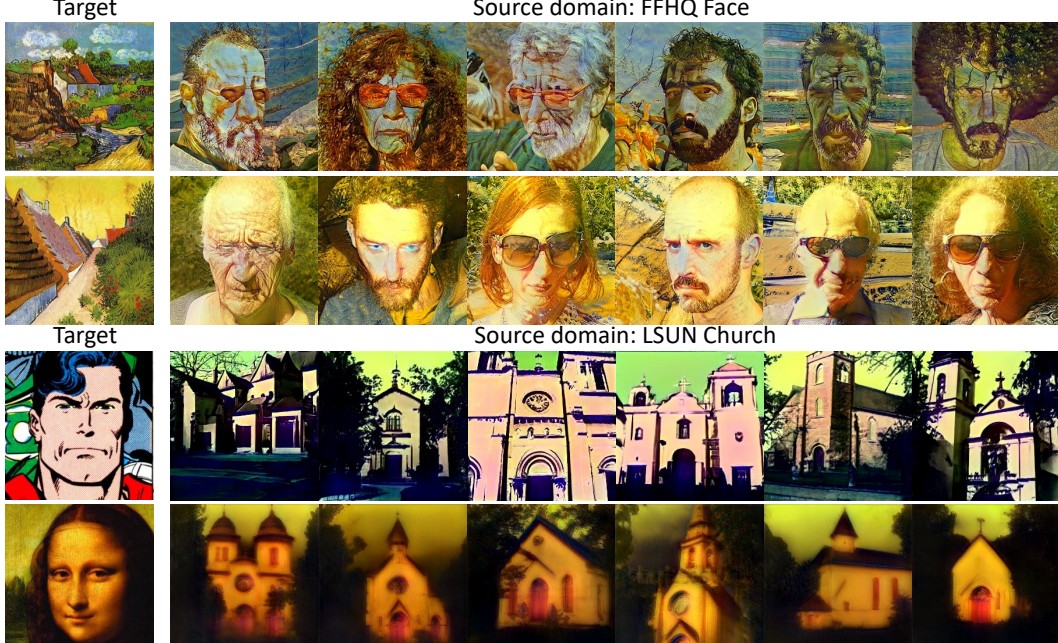

Figure 5: **Cross-domain adaptation**, where GenDA manages to transfer the character of an out-of-domain target (first column) to the source domain.

## 3.3 CROSS-DOMAIN ADAPTATION

In this part, we study the adaptation on unrelated source and target domains. Specifically, Van Gogh's houses, Superman and Mona Lisa serve as the target for a face and church source model respectively. Considering the motivation that we aim at reusing the prior knowledge (*i.e.*, variation factors) by freezing the parameters, the synthesis after adaptation is supposed to share similar visual concepts. That is, a face model would still produce faces no matter what the target image is. Fig. 5 suggests that the source models remain to produce the corresponding content. More importantly, the color scheme and painting styles are also transferred. For example, the red roof at the first row renders the red glasses, the yellow sky at the second row draws the front head in yellow. The blue hair, green background and the shadows of Superman are well adapted to church. The painting style of Mona Lisa is also transferred.

Obviously, the shared attributes between face and church are quite rare. Therefore, GenDA pours more attention to the variation factors like color scheme, texture and painting styles which could be directly transferred across unrelated domains. From this perspective, our GenDA might enable a new alternative for neural style transfer task that aims at transferring the styles of a given image. But more generally, our GenDA is able to transfer more high-level attributes like gender, age, and sunglasses while the technique of style transfer might fail, which might be of benefit to art creation.

## 4    RELATED WORK AND LIMITATIONS

**Training Generative Models with Limited Data.**    Many attempts have been taken to train a generative model on limited data. For one thing, some of the prior approaches proposed to leverage the data augmentation to prevent the discriminator from overfitting. Specifically, Zhang & Khoreva (2019) introduced a type of progressive augmentations. Zhao et al. (2020d) investigated the effects of various augmentations during the training. Theoretical analysis was conducted by Tran et al. (2021) for several data augmentations. Zhao et al. (2020b) proposed to apply the augmentations to both the real and synthesized images in a differentiable manner. Karras et al. (2020a) designed an adaptive discriminator augmentation that does not leak to stabilize the training process. For another, multiple regularizers were also introduced to provide extra supervision. For instance, Zhao et al. (2020c) involved the consistency regularization for GANs which shows competitive performances with limited data. Yang et al. (2021a) incorporated contrastive learning as an extra task to improve the data efficiency. Karras et al. (2020a) also pointed out that decreasing the number of parameters of generators and introducing the dropout (Srivastava et al., 2014) could alleviate the overfitting problems. Shaham et al. (2019) and Sushko et al. (2021) designed different frameworks which learn from a single natural image or video. However, when the number of available images is less than 10, they usually lead to unsatisfying diversity. Different from these works which learn from scratch, we focus on the generative domain adaptation, a practical alternative that first pre-trains a source model on the large-scale dataset and then transfers it on a target domain with the one-shot image.

**Few-shot Generative Domain Adaptation.**    Generative domain adaptation has attracted a considerable number of interests due to its practical importance. Wang et al. (2018) proposed to use the same objective for adaptation. Noguchi & Harada (2019) fine-tuned the batch statistics merely for the few-shot adaption. Wang et al. (2020) transformed the original latent space and tuned the entire parameters for the target domain. Mo et al. (2020) froze the lower-level representations of the discriminator to prevent overfitting. Zhao et al. (2020a) revealed that low-level filters of both the generator and discriminator can be transferred via a new adaptive filter modulation. Li et al. (2020) penalized certain weights identified by Fisher information. Robb et al. (2020) learned to adapt the singular values of the pre-trained weights while freezing the corresponding singular vectors. Ojha et al. (2021) proposed the cross-domain consistency as a regularization to maintain the diversity. Different from prior work, we aim at reusing the learned knowledge on the source domain and thus identify and adapt the representative attributes, which enables the one-shot domain adaptation and outperforms other alternatives by a clear margin under the general few-shot settings.

**Limitations.**    Despite the state-of-the-art performances on both one-shot and few-shot generative domain adaptation, our proposed GenDA still has some limitations. For example, the rationale behind GenDA is to reuse the prior knowledge learned by the source GAN model, which hinders it from transferring a model to a completely different domain. As suggested in Fig. 5, when we adapt a church model regarding a face image, the outputs are still churches but not faces. This implies that our method would fail when the inter-subclass variations are huge. Such a property is a silver lining, depending on the practical application. A second limitation is that our current design treats all characters of the reference image as a whole. Taking the first row of Fig. 3 as an example, the sunglasses, skin color, and background are transferred simultaneously. It is hard to accurately transfer some particular attributes. However, it is indeed possible to use some auxiliary samples to help define a common attribute, as shown in Fig. 4. Besides, our GenDA also relies on the layer-wise stochasticity involved in the generator structure. Concretely, in our base model, StyleGAN2 (Karras et al., 2020b), the latent code is fed into all convolutional layers instead of the first layer only. Without the layer-wise design, the supervision will be hard to back-propagate to the attribute adaptor given a deep synthesis network. Fortunately, however, such a design is commonly adopted by the state-of-the-art GANs (Karras et al., 2019; 2020b; Brock et al., 2018).

## 5    CONCLUSION

In this work, we propose **GenDA** for one-shot generative domain adaptation. We introduce two lightweight modules, *i.e.*, an *attribute adaptor* and an *attribute classifier*, to the fixed generator and discriminator respectively. By efficiently learning these two modules, we manage to reuse the prior knowledge and hence enable one-shot transfer with impressively high diversity. Our method demonstrates substantial improvements over existing baselines in a wide range of settings.

## ETHICS STATEMENT

Training a generative model with limited data is a fundamental and practical problem. This work significantly advances this field by using as few as only one image for domain adaptation. Such an improvement is of great importance to the community. Firstly, it pushes the data efficiency in GAN training to the limit and demonstrates an early success. We believe more and more methods will be proposed to keep improving the training efficiency along this direction. Secondly, our approach is based on reusing the prior knowledge of a pre-trained GAN model. It provides insights on how to better utilize the pre-trained generative model for various downstream tasks (like semantic manipulation and super resolution), which is a widely studied research area. Thirdly, this work also enables many potential real-world applications, like data augmentation (to solve the long-tail problem), entertainment, creative design, artistic imitation, *etc*. However, every coin has two sides. Our method considerably reduces the cost of producing fake data, in terms of using less data, less time, and fewer computing resources. But we believe that its academic contribution and its positive impact on the industrial algorithm are more valuable, and that the negative impact will be lowered along with the development of deep fake detector.

## REPRODUCIBILITY STATEMENT

The base model used in this work is StyleGAN2 (Karras et al., 2020b), with official implementation stylegan2-ada-pytorch. The large-scale source datasets we use are FFHQ and LSUN Church. For one-shot domain adaptation, the reference image can be easily customized by the users, either from online or from other datasets, since our approach only requires one training sample. The implementation and training details are described at the beginning of Sec. 3, which are shared by all experiments. Besides, all experiments are conducted with 8 GPUs. The evaluation metrics, including FID (Heusel et al., 2017), precision and recall (Kynkäänniemi et al., 2019), also follow their official implementation, FID and improved-precision-and-recall-metric. To ensure the reproducibility, the code and models will be made publicly available.

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

# APPENDIX

This appendix is organized as follows. Sec. A conducts ablation study on the proposed approach, including the *attribute adaptor*, the *attribute classifier*, the *diversity-constraint strategy*, and the hyper-parameter $\beta$ in Eq. (5). Sec. B analyzes the transformation (*i.e.*, the learnable vector $\mathbf{a}$ in Eq. (3)) for attribute adaptation, which provides insights on how our algorithm works. Sec. C shows how the latent representations are transformed after adaptation. Sec. D presents the additional comparison with the style transfer method and investigates the visual relation between and after adaptation. Sec. E performs latent interpolation with the models adapted on the target domain, which suggests that our GenDA does not harm the learned latent structure.

Table A1: **Ablation study on the components proposed in GenDA**, including the attribute adaptor (AA), the attribute classifier (AC), and the diversity-constraint strategy (DC). Evaluation metrics include FID (lower is better), precision (higher means better quality), and recall (higher means higher diversity).

| AA | AC | DC | FID | Prec. | Recall | FID | Prec. | Recall | FID | Prec. | Recall |
|----|----|----|-----|-------|--------|-----|-------|--------|-----|-------|--------|
|    |    |    | 109.87 | 0.80 | 0.000 | 53.06 | 0.72 | 0.003 | 39.96 | 0.70 | 0.024 |
| ✓  |    |    | 53.59 | 0.57 | 0.064 | 28.35 | 0.72 | 0.333 | 37.68 | 0.30 | 0.034 |
| ✓  | ✓  |    | 50.37 | 0.62 | 0.106 | 25.29 | 0.71 | 0.349 | 33.29 | 0.39 | 0.089 |
| ✓  | ✓  | ✓  | 47.05 | 0.71 | 0.065 | 22.62 | 0.78 | 0.204 | 31.97 | 0.57 | 0.076 |
|    |    |    | (a) Babies | | | (b) Sunglasses | | | (c) Sketches | | |

# A ABLATION STUDY

**Component Ablation.** In order to further investigate the role of each proposed component (*i.e.*, attribute adaptor, attribute classifier and truncation strategy), we conduct comprehensive ablation studies on 10-shot adaptation. In particular, we choose FreezeD (Mo et al., 2020) as our baseline model which freezes the lower features of the discriminator while fine-tunes the rest of the discriminator and the entire generator. Moreover, we also apply adaptive discriminator augmentation strategy (ADA) on it since data augmentations usually help alleviate the overfitting problem.

As is shown in Tab. A1, the baseline alternative could achieve the satisfying synthesis quality but worse diversity, suggesting that it might overfit the training shot significantly. By involving the attribute adaptor, the baseline is substantially improved, which already outperforms prior approaches. Meanwhile, the boosted diversity also demonstrates our motivation that we could reuse the variation factors of the source model and hence maintain the synthesis diversity to some extent.

Table A2: **Ablation study on architecture designs.** A two-layer multi-layer perceptron (MLP) is introduced as the heavy version of the attribute adaptor and the attribute classifier, respectively.

| Models | Babies | Sunglasses | Sketches |
|---|---|---|---|
| Heavy attribute adaptor | 290.22 | 262.32 | 114.25 |
| Heavy attribute classifier | 132.14 | 115.88 | 109.23 |
| Lightweight design | **80.16** | **44.96** | **87.55** |

Table A3: **Ablation study on the strength of diversity-constraint strategy**, which is noted as $\beta$ in Eq. (5). Here, a smaller $\beta$ denotes a stronger diversity constraint. Evaluation metrics include FID (lower is better), precision (higher means better quality), and recall (higher means higher diversity).

| $\beta$ | FID | Precision | Recall |
|---|---|---|---|
| 0.5 | 24.87 | 0.82 | 0.163 |
| 0.7 | 22.62 | 0.78 | 0.204 |
| 0.9 | 24.32 | 0.73 | 0.269 |
| 1.0 | 25.29 | 0.71 | 0.349 |

However, the quality (*i.e.*, precision) on babies and sketches becomes worse. After being equipped with the attribute classifier, the overall measurement is further enhanced. Together with the diversity-constraint strategy, our GenDA could facilitate the quality to the baseline level but achieve better diversity, leading to the new state-of-the-art few-shot adaptation performances.

**Architectural Ablation.** To further study the effect of the lightweight design of the attribute adaptor and the attribute classifier, we conduct experiments of replacing the lightweight module with two-layer multi0layer perceptron (MLP), which has a higher learning capacity. Tab. A2 presents the results. Obviously, with either a heavier attribute adaptor or a heavier attribute classifier, the synthesis performance drops significantly. This might be caused by the overfitting problem since we only have one training image under our one-shot setting. It verifies the effectiveness and necessity of our lightweight design for the challenging one-shot generative domain adaptation task.

**Hyper-parameter Ablation.** We also conduct the ablation of hyper-parameter $\beta$ in Eq. (5). Specifically, we train our model with 10-shot sunglasses images and maintain the evaluation metrics used in Sec. 3.2. As is shown in Tab. A3, when increasing the value of $\beta$, the synthesis quality becomes worse while the diversity is enhanced. Noticeably, when the $\beta$ is less than 0.5, the training diverges. Therefore, we choose 0.7 in all experiments for its overall performance.

# B  ANALYSIS ON TRANSFORMATION VECTOR

We also visualize the learned attributes via Principal components analysis (PCA). Concretely, we first choose 15 shots from 3 domains (*i.e.*, babies, sunglasses and sketches). The corresponding models are trained on them and 15 attribute vectors in Eq. (3) are collected to perform PCA. Fig. A1 presents the PCA results. Obviously, the attribute vectors of different shots but the same domain assemble closely, while there are obvious decision boundaries across domains. Such interpretation happens to match our motivation that the attribute adaptor could capture the representative attributes and make the corresponding adjustment for a source model to multiple target domains.

# C  ANALYSIS ON TRANSFORMED LATENT REPRESENTATION

We also visualize how the latent representations change before and after adaptation, with the help of PCA. Specifically, we sample $2K$ latent codes from the latent space after adaptation for each model used in Fig. 4 and Fig. 5. Fig. A2 suggests that the distributions after adapting the source model to three different target domains are clearly separable. Also, when using more than one reference image, which have common attributes, from the target domain, the transformed representations tend to locate at the overlapped region between the representations learned from each reference image independently. Fig. A3 shows the latent representation change after cross-domain adaptation, where the latent distribution of the source domain is successfully shifted to the target one.

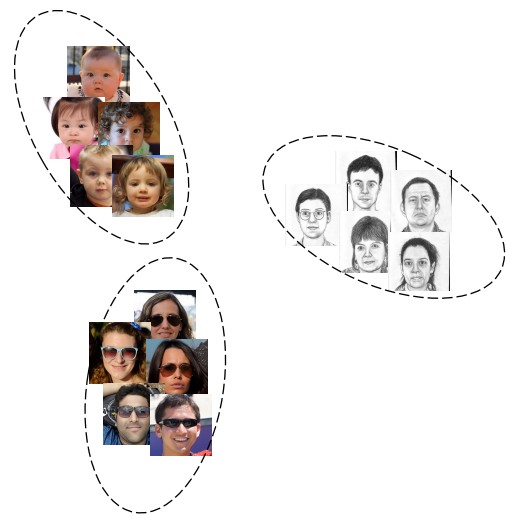

Figure A1: **Analysis on the transformation vector for domain adaptation**, which is noted as **a** in Eq. (3). We launch 15 one-shot adaptation experiments with 5 kids, 5 sunglasses, and 5 sketches as the target, respectively. After the model converges, we collect the transformation vector learned by the *attribute adaptor*, and then perform PCA on all 15 vectors. We can tell that our GenDA tends to learn similar transformation regarding the reference with similar characters. The 15 training samples are visualized together with the PCA results.

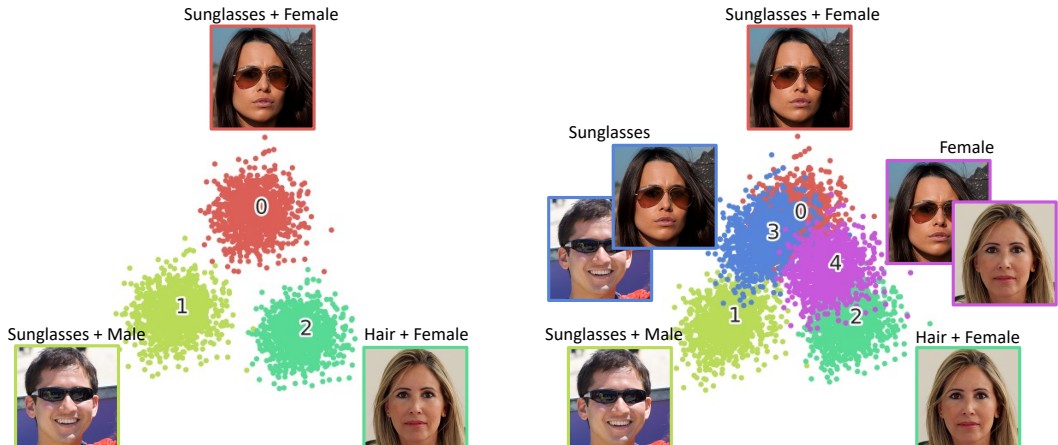

Figure A2: **Visualization of transformed latent spaces after domain adaptation.** Each cluster corresponds to a particular training set. Left: The latent spaces learned with different reference images are clearly separable. Right: When we use two reference images, which have common attributes, as a combined training set, the transformed representations tend to locate at the overlapped region between the representations learned from each reference image independently.

## D  ADDITIONAL RESULTS

**Comparison with the style transfer task.** As discussed in Sec. 3, the proposed approach might be an alternative for style transfer. Here, we qualitatively compare our approach with the state-of-the-art method, SWAG (Wang et al., 2021) regarding the style transfer task. The comparison results are included in Fig. A4. From the perspective of the color transfer, SWAG (Wang et al., 2021) does better since it could successfully transfer the representative color from the target image to the synthesis. For example, the green of the vegetation and the pink of the skin appears in the transferred samples. However, it also reveal its weakness that the transfer is inconsistent. Specifically, the wall or the sky in church synthesis is in pink while our method pictures them in the same color. This

| Source Domain | Target Domain | Synthesis After Adaptation | Representation Change |
|---|---|---|---|

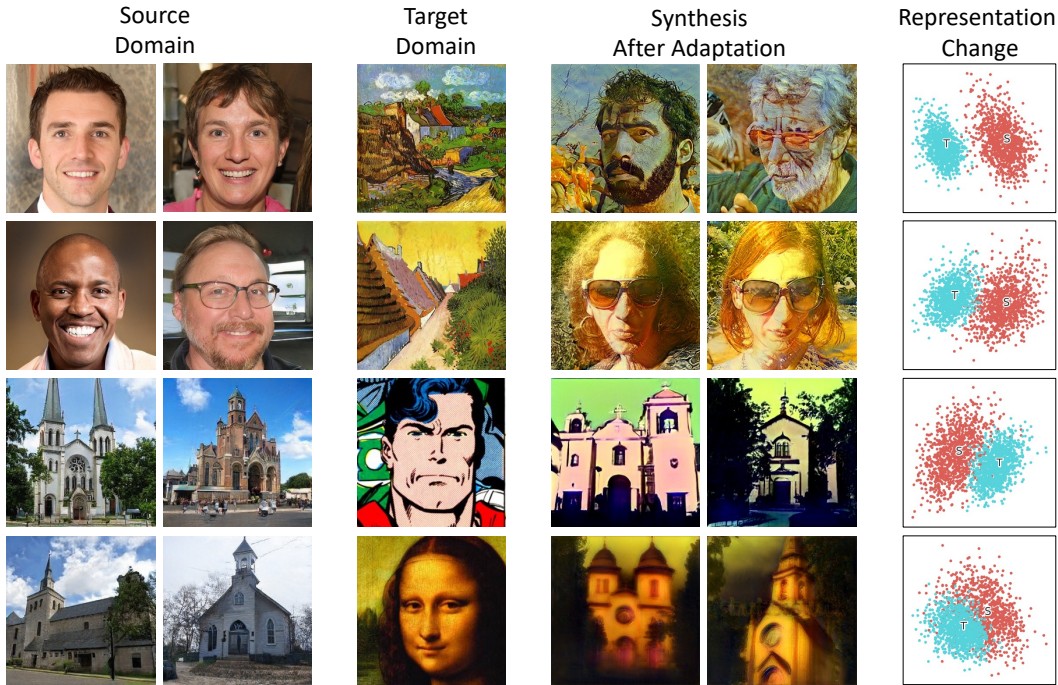

Figure A3: **Latent representation comparison before and after adaptation.** The latent space is clearly pushed from the **red** cluster to the **blue** cluster when there exists an obvious gap between the source and the target domains.

implies that the style transfer based methods could have no semantic concepts regarding to the content, leading to such inconsistency and obvious artifacts. Therefore, it naturally cannot transfer higher-level attributes like sunglasses while ours could.

**The visual relation before and after adaptation.** A byproduct of our proposed GenDA is that we manage to inherit some characteristics from the source model in the process of domain adaptation. Here, we take the images synthesized by the source generator as the references, and feed the adapted generator with the same latent codes. In this way, we can check how the synthesis varies before and after adaptation. Fig. A5 and Fig. A6 correspond to the settings of Fig. 4 and Fig. 5, respectively. We can tell that many attributes are maintained in the adapting process, verifying our motivation, which is that the most of variation factors could be reused in the task of generative domain adaptation.

# E INTERPOLATION OF LATENT VECTORS

Here, we perform the semantic interpolation to evaluate the tuned generator. Typically, the interpolation at the latent space (Radford et al., 2015) could interpolate the semantics of two images, which is widely used to measure the diversity of generative models. Fig. A7 and Fig. A8 present the semantic interpolation of face and church model respectively. In particular, all models are fine-tuned with one shot image. Obviously, satisfying interpolation *i.e.*, smoothing change is clearly observed, even if the model is fine-tuned with one shot image.

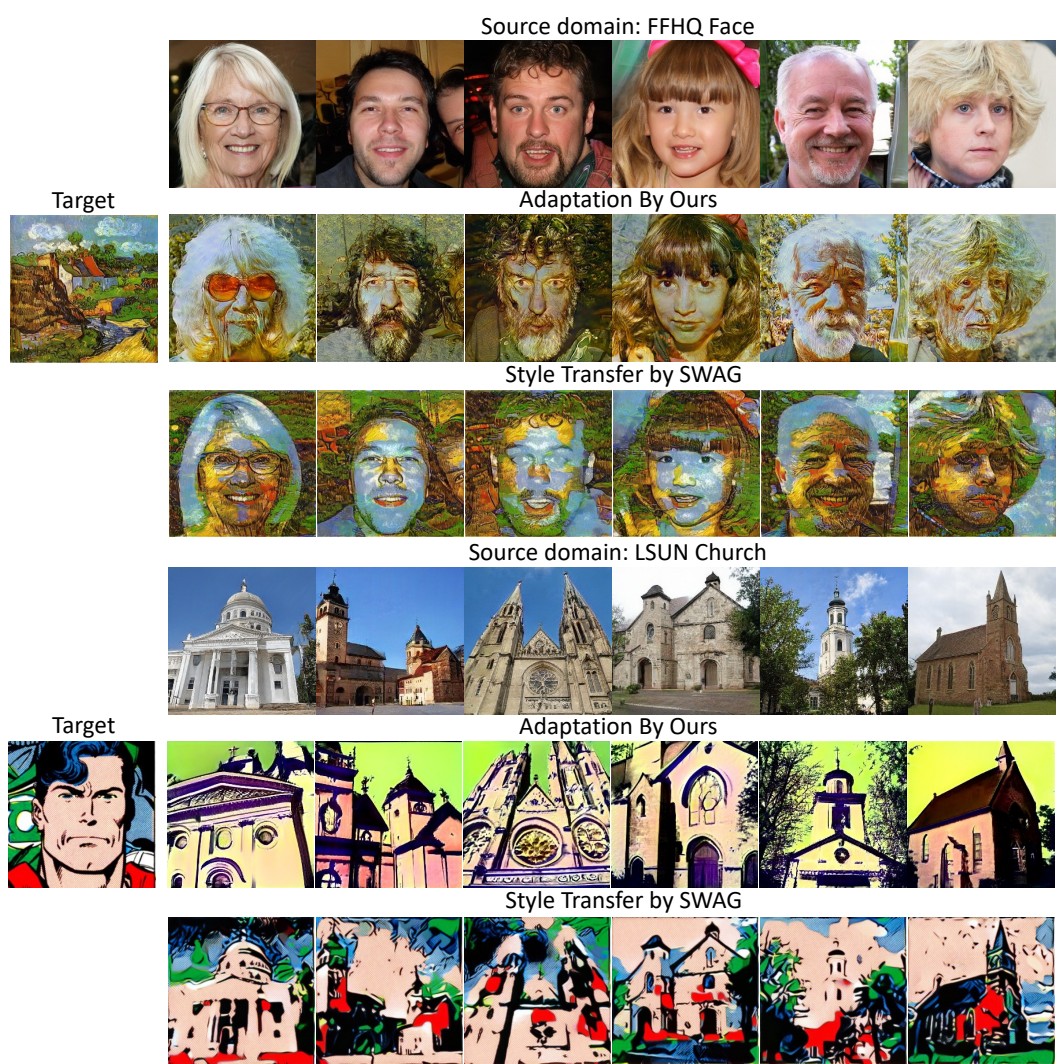

Figure A4: **Comparison with the style transfer task.** GenDA could transfer the characteristic of the target image to the source model in more harmonious way, while SWAG (Wang et al., 2021) transfers the color inconsistently, leading to obvious artifacts.

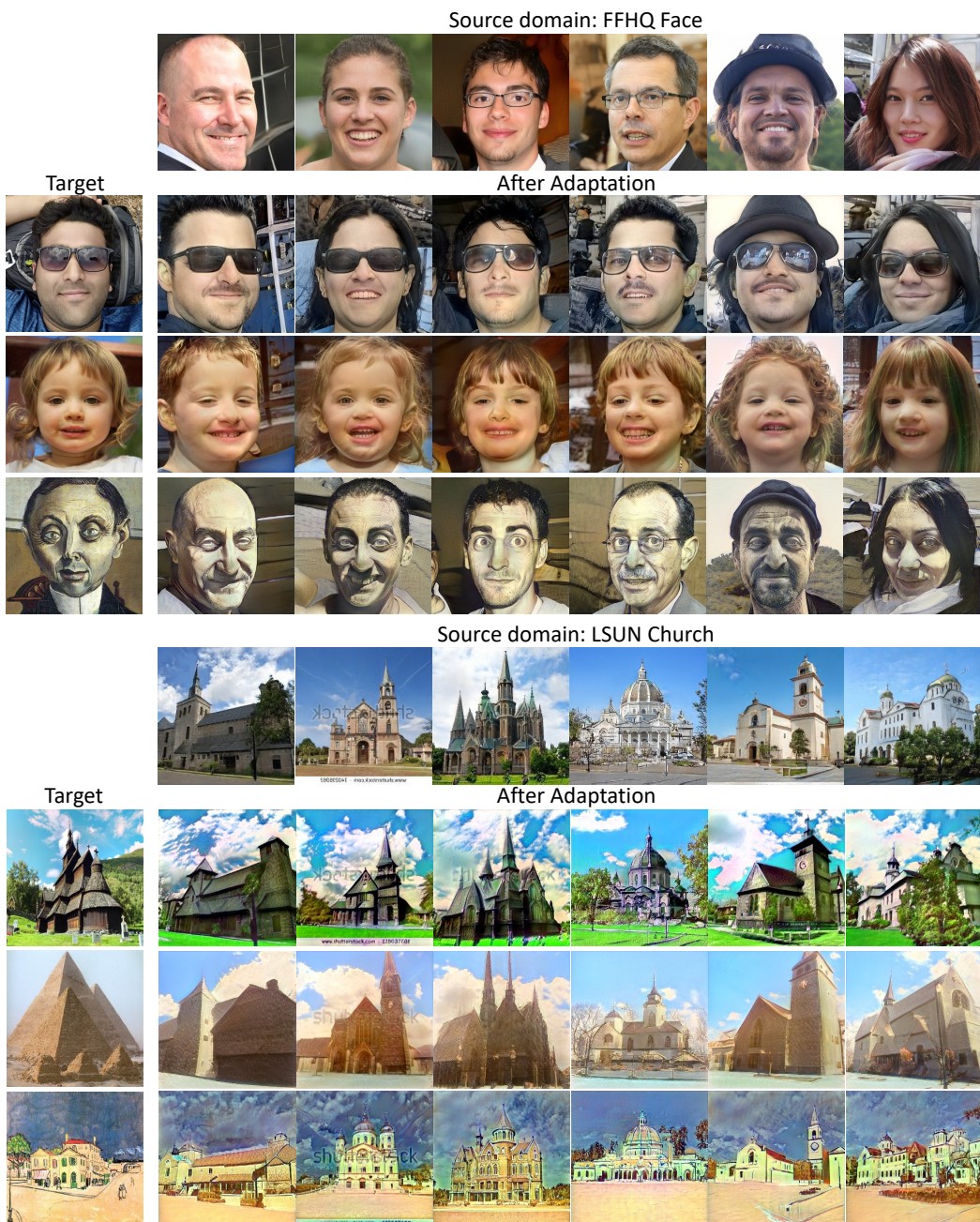

Figure A5: **Visual relation of one-shot adaptation with different target samples.** Here, for each model (*i.e.*, faces and churches), each column is generated with the same latent code. Our GenDA is able to transfer the most distinguishable attributes of the target domain, but almost maintains other semantics (*e.g.*, the face pose and the church shape).

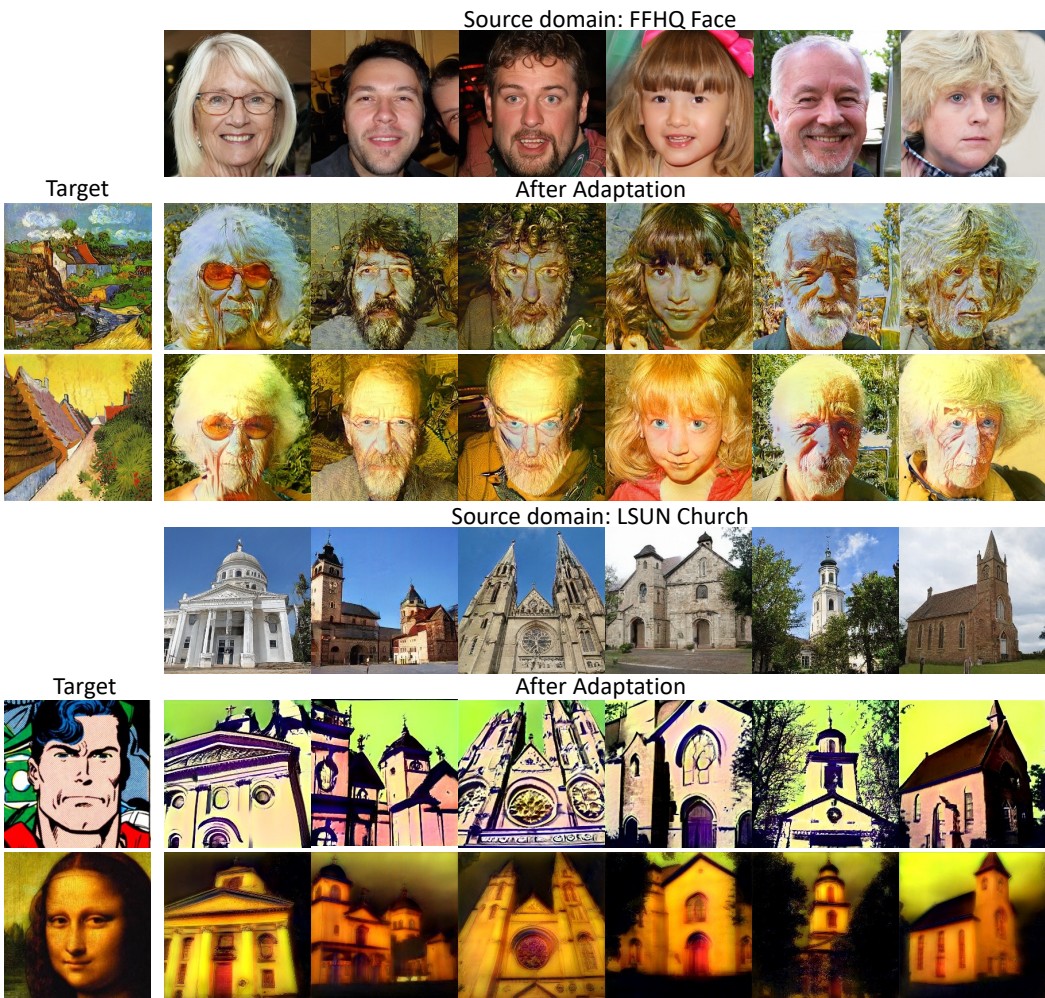

Figure A6: **Visual relation of cross-domain adaptation.** Here, for each model (*i.e.*, faces and churches), each column is generated with the same latent code. We can tell that, under the cross-domain setting, our GenDA can still inherit some characteristics from the source model (*e.g.*, the face pose and the church shape).

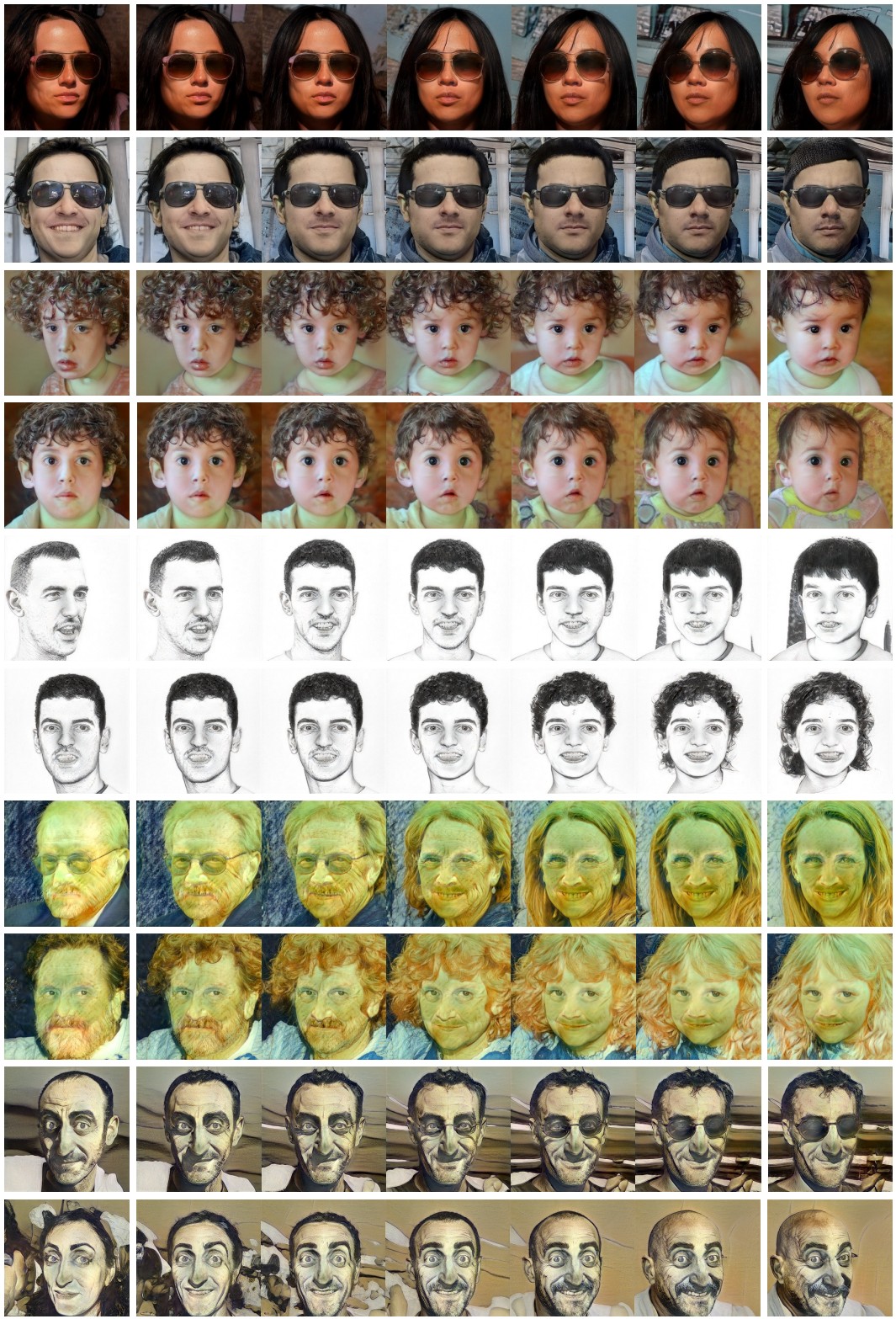

Figure A7: **Semantic interpolation** with the model pre-trained on FFHQ human faces (Karras et al., 2019). The first and last columns are the synthesized images with two different latent codes after one-shot domain adaption, while the remaining columns are the outputs by linearly interpolating the two codes. All intermediate results are still high-quality faces, with gradually varying semantics (*e.g.*, the gender and the face pose).

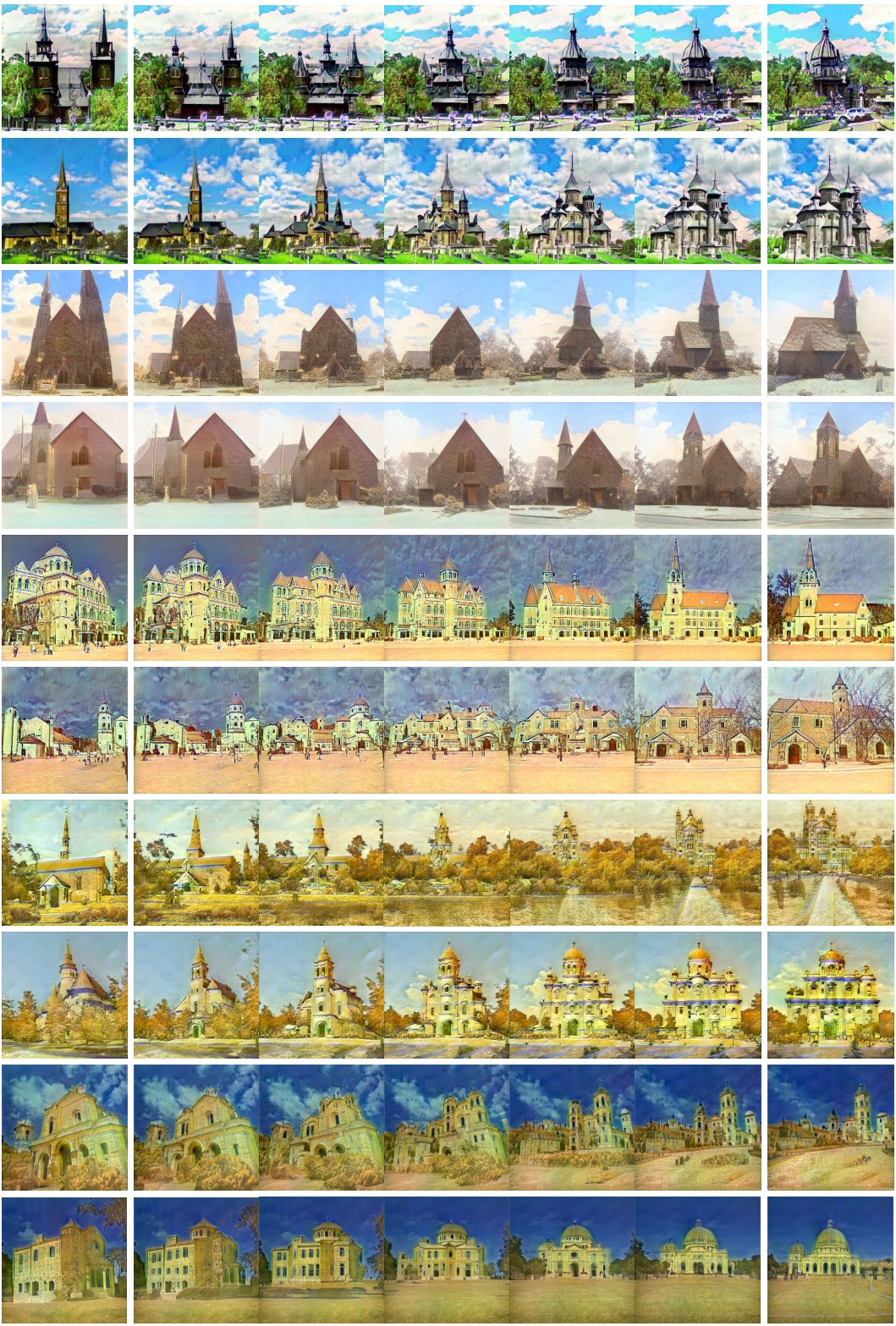

Figure A8: **Semantic interpolation** with the model pre-trained on LSUN Churches (Yu et al., 2015). The first and last columns are the synthesized images with two different latent codes after one-shot domain adaption, while the remaining columns are the outputs by linearly interpolating the two codes. All intermediate results are still high-quality churches, with gradually varying semantics (*e.g.*, the church shape and the cloud in the sky).

