# OpenReview forum: "One-Shot Generative Domain Adaptation"
_ICLR.cc/2022/Conference — ICLR 2022 Submitted_

### Official Review · Reviewer_CemL · 2021-11-01

**Correctness:** 3
**Technical Novelty And Significance:** 2
**Empirical Novelty And Significance:** 2
**Recommendation:** 5
**Confidence:** 5

**Main Review:**

Pros:
1. The one-shot image generation is a novel and interesting task.

2. The overall idea and the proposed pipeline towards addressing it are easy to follow.

3. In the experimental, the claim about one-shot attribute-related results is being met.

Cons:

While the authors aim to address a challenging and novel task, I believe some parts need more clarification (even after considering the supplementary material):

1. The biggest weakness is that the proposed method has limited novelty. While the authors propose a stacked pipeline to address the quality and diversity, the key contribution they made is unclear.

a. The z+/w/w+/s space analysis and adaption has been widely conducted in the latest works [r1, r2, r3]. What are the differences between the proposed adaptor and these prior works? Why the proposed adaptor would like to perform better?

b. Related to the above, the attribute classifier has been used in StyleFlow [r2]. Why the proposed one is better? In addition, if I understand correctly, the attribute classifier only judges the output is real or fake, instead of predicting attribute labels, because some examples in Figures 2 and 3 should not have corresponding labels. If this classifier just outputs real or fake labels, why not just fine-tuning the final layer of the original discriminator?

c. I cannot buy the novelty of reusing truncation trick for diversity-constraint strategy. As mentioned by the authors, this trick is a normal one in the current generation code. The authors did not provide a new direction to sell this strategy.

2. The impressive results in Figures 4 and 5 are interesting, but the analysis is not clear enough. It would like to be better to visualize the learning representation after the adaptor as in AgileGAN [r3].

a.  The results in the final two rows of Figure 4 are quite interesting. I guess the learned w space has been pushed to a narrow space (the final one will be more narrow). If the authors can visualize the learning representation distribution, it will be clearer to demonstrate such mapping in the representation domain.

b. A similar situation is in Figure 5.  It is interesting to use the one-shot pipeline for style translation. As the pre-trained G and D have not been updated, I guess the learned w space has been pushed into a different distribution to the original w space. How to visualize such change and then demonstrate the swap of learning representation would be significant.

[r1] Wu, Z., Lischinski, D., & Shechtman, E. (2021). Stylespace analysis: Disentangled controls for stylegan image generation. In Proceedings of the IEEE/CVF Conference on Computer Vision and Pattern Recognition (pp. 12863-12872).
[r2] Abdal, R., Zhu, P., Mitra, N. J., & Wonka, P. (2021). Styleflow: Attribute-conditioned exploration of stylegan-generated images using conditional continuous normalizing flows. ACM Transactions on Graphics (TOG), 40(3), 1-21.
[r3] Song, G., Luo, L., Liu, J., Ma, W. C., Lai, C., Zheng, C., & Cham, T. J. (2021). AgileGAN: stylizing portraits by inversion-consistent transfer learning. ACM Transactions on Graphics (TOG), 40(4), 1-13.

**Summary Of The Paper:**

In this work, the authors propose a framework for one-shot generation, which aims to tame a new GAN using a pre-trained GAN model as well as one target image. Here, the authors present three strategies to ensure the quality and diversity of generated images: introducing an attribute adaptor to map the latent space to the target attribute; presenting an attribute classifier in the discriminator, and applying the truncation trick for the training. Experiments are conducted on the face and building datasets.

**Summary Of The Review:**

My preliminary rating for this submission is marginal reject. While I like this novel task that aims to generate diverse images with high quality, the key novelty and contribution are unclear explained to the prior related works. In addition, interesting experimental results are provided, but they are not well analyzed in the representation domain. The realistically visual results cannot demonstrate the contribution perfectly.

---

> ### Author Response · Authors · 2021-11-17
> **Response to Reviewer CemL**
>
> **Q1. The biggest weakness is that the proposed method has limited novelty.**
>
> The main contribution of this paper is to **enable the task of one-shot generative domain adaptation**. Previous work barely achieves the goal of adapting a GAN model with **only one image**. Recall that generative domain adaptation is **a clearly different task from facial editing**, which is targeted by the mentioned work [r1] and [r2].
>
> **Q2. What are the differences between the proposed adaptor and these prior works? Why the proposed adaptor perform better?**
>
> Although these works also transform the latent space to some extent, the targets are totally different. For the purposes of disentangled controls, [r1] analyzes the latent spaces and leverages a pre-trained classifier to identify certains channels that control a specific attribute. In order to support conditional sampling, [r2] uses conditional continuous normalizing flows to learn the relationship between two latent spaces. Besides, [r3] proposes a framework that can generate high quality stylistic portraits with a hierarchical variational encoder that ensures the inversion consistency. These works are clearly in a different scope from our work. Please refer to **Q1**.
>
> Concretely, our attribute adaptor aims at adapting the most representative characters of the target domain to the source generator. In this way, through randomly sampling from the latent space, the adapted generator can always synthesize images that possess the characters of the target domain. Here, the most representative characters are identified in the data-driven manner, without the help of any pre-training. Moreover, our adaptor is designed as a lightweight transformation module, instead of continuous normalizing flows ([r2]) or a hierarchical variational encoder ([r3]). Such a design makes our adaptor more friendly to *one-shot* generative domain adaptation, where there is only one training sample. Experimental results in Tab. A2 also suggest that a heavier network might cause severe performance degradation.
>
> **Q3. Why the proposed attribute classifier is better? Why not just fine-tuning the final layer of the original discriminator?**
>
> The attribute classifier we refer to is fundamentally different from the attribute classifier used in the mentioned literature. Specifically, the latter pours more attention to specific attributes (*e.g.*, gender, age, and hair) and is mostly pre-trained in the supervised manner. Instead, our attribute classifier aims at distinguishing the domain-specific attributes of the target domain from those of the source domain, and gets updated from scratch in the adversarial learning.
>
> Our experimental results in Tab. A1 suggest that incorporating a new classifier (the “AA + AC” row) outperforms directly fine-tuning the final layer of the original discriminator (the “AA” row). It implies that there exists a gap between (1) the classification of real and fake images in the source domain, and (2) the domain-specific attribute classification in our approach.
>
> **Q4. I cannot buy the novelty of reusing truncation trick for diversity-constraint strategy.**
>
> Previous approaches apply the truncation trick to improve the quality **at the inference stage**. However, it is the first time to leverage it at the training stage as a diversity-constraint strategy. This is motivated by “there is only one reference image from the target domain, but the source generator can produce high-diverse images”. Such a truncation strategy helps lessen the diversity gap between the domains before and after adaptation. Tab. A1 also demonstrates the effectiveness of this strategy.
>
> **Q5. If the authors can visualize the learning representation distribution of Figure 4, it will be clearer to demonstrate such mapping in the representation domain.**
>
> Thanks. As suggested, we include the visualization of latent representations in Fig. A2 of *the revised version*. Obviously, the latent representations are clearly pushed away given different reference images as the targets. Also, when using more than one reference image from the target domain, the transformed representations tend to locate at the overlapped region between the representations learned from each reference image independently.
>
> **Q6. How to visualize the representation change in Figure 5 and then demonstrate the swap of learning representation would be significant.**
>
> We include the representation change after adaptation in Fig. A3 of *the revised version*. Apparently, the latent representations are successfully shifted compared with the original ones.

---

> > ### Author Response · Authors · 2021-11-23
> > **Any further comment?**
> >
> > Dear Reviewer CemL,
> >
> > Thank you so much for your effort in reviewing this submission! We have tried our best to address your concerns above the reply. Would you mind taking a look and letting us know what you think?
> >
> > Feel free to let us know if there is anything unclear or so. We are happy to clarify them.
> >
> > Best,
> > Authors

---

> > > ### Comment · Reviewer_CemL · 2021-11-23
> > > **Thanks for the authors' reply**
> > >
> > > Thanks for the author's reply. Part of my concerns are solved, especially for the feature visualization in Q5 and Q6. The added discussions are interesting. However, I still feel that the proposed method lacks novelty.
> > > - Q1: As claimed in the comments, I fully agree that one-shot generation is a novel and interesting task. However, the two modifications and one trick are simply designed, yet why they do work well is not clearly demonstrated in the paper. Zh5e "The adaptor does not seem powerful enough to match the complexity of the task".
> > > - Q3: I noted that result, but just wonder why? Does it mean the final layer in the pre-trained discriminator is much important in such a setting? We must use the whole pre-trained discriminator for the classifier, instead of different layers used in the perceptual loss.
> > > - Q4: I guess this is a training trick. I cannot buy this as a new contribution to the community.

---

> > > > ### Author Response · Authors · 2021-11-23
> > > > **Response to further comments**
> > > >
> > > > Thanks for liking the novel and interesting task studied in this work, which is also appreciated by the other two reviewers.
> > > >
> > > > First, we would like to reaffirm that, under such a challenging yet useful setting (*i.e.*, adapting a generative model **with only one reference image**), our approach **makes it work for the first time** to the best of our knowledge. From this point of view, it should already be considered as a major contribution and **of great interest to the community**. It **provides a valid starting point for this research direction**. Here, by saying "make it work", we provide **sufficient qualitative and quantitative proof regarding both the quality and the diversity criteria**. Also, as suggested, we visualize the latent representations before and after adaptation in Fig. A2. Our approach indeed transfers the latent space successfully.
> > > >
> > > > Second, we argue that **novelty does not necessarily mean a complex system**, and that an algorithm should not be blamed for its simplicity. Instead, **simple but efficient** approaches should be encouraged because of the **easy implementation, good reproducibility, and strong generalization ability**. For example, ResNet (using skip-connection to solve the gradient vanishing problem) and MoCo (leveraging a memory bank to increase the number of negative samples for contrastive learning) are both this kind of methods. Our GenDA is **efficient** (only a few parameters are tuned) but far more **effective** than existing alternatives (see comparisons in Tab. 1 and Tab. 2). As a result, we manage to transfer a GAN model **within a few minutes**, which is another appealing advantage of our approach.
> > > >
> > > > Third, about the questions "why they do work well is not clearly demonstrated in the paper" and "I noted that result, but just wonder why". We have conducted comprehensive ablation studies and shown the results in **Appendix**.
> > > >
> > > >    - Tab. A1 suggests that **without the newly introduced attribute adaptor (AA) or attribute classifier (AC), the performance remains poor**. The reasons are already discussed in Sec. 2.2 and the rebuttal. Here, we carefully pick some discussions.
> > > >       - On one hand, "the generator focuses on transferring the most distinguishable characters of the only reference, **instead of learning the common variation factors repeatedly**".
> > > >       - On the other hand, "the generated images before and after domain adaptation are expected to share most variation factors, therefore, **the knowledge learned by the discriminator in its pretraining could be also reused**".
> > > >       - Also, the comparison between the first row and the third row of Tab. A1 suggests that **tuning the whole discriminator achieves worth performance than our GenDA**. The comparison between the second row and the third row of Tab. A1 suggests that **without re-initializing AC also achieves worth performance than our GenDA**.
> > > >       - "Why the last layer is more important" is explained in the rebuttal, as "**there exists a gap** between (1) the classification of real and fake images in the source domain, and (2) the domain-specific attribute classification in our approach".
> > > >
> > > >    - Tab. A2 suggests that **with a heavier attribute adaptor or attribute classifier, the performance also becomes worth**. That is because, under our one-shot setting, there is only one training sample, which may easily cause the overfitting problem. Our approach **works just because of its efficient design**.
> > > >
> > > > Hope that the above discussions address your concerns. Thanks for your effort and suggestions again.

---

### Official Review · Reviewer_NUUH · 2021-11-01

**Correctness:** 4
**Technical Novelty And Significance:** 3
**Empirical Novelty And Significance:** 4
**Recommendation:** 8
**Confidence:** 5

**Main Review:**

Strengths:
* This paper tackles a challenging domain adaptation problem which is very interesting.
* This paper demonstrates convincing qualitative comparisons (e.g., realism and diversity) to the existing efforts including Mo et al., 2020 and Ojha et al. 2021.
* This paper also shows comparable and improved quantitative results to the state-of-the-art methods in the case of a one-shot and ten-shot adaptation, respectively.

Weaknesses:
* This paper discovered an interesting phenomenon that a lightweight attribute adaptor could generate surprisingly-good domain adaptation results. Although it has been explained in the paper, the reviewer would appreciate it if some sort of ablation study (both qualitative and quantitative results) can be included in the paper (at least in the supplementary material of the final version). For example, what happens if we use different architecture or operators for the attribute adaptor.  The same rule applies to the attribute classifier (though it might be overfitting).
* At a higher level, this paper reminds me of the Exemplar SVM published 10 years ago. The conclusion in the paper is that it is often sufficient to obtain a good decision boundary if you have one positive example and many negative examples. This is very similar to the attribute classifier proposed in the paper. The reviewer would suggest incorporating such discussions in the final version of the paper.
* The limitation of the paper is not clearly stated. For example, the reviewer would like to know if the one-shot learning method applies to the “car -> abandoned car” application studied in the Ojha et al., 2021. It seems to the reviewer that the proposed method could fail at “car -> truck” application when the inter-subclass variations are huge. Please comment on this in the rebuttal.
* Furthermore, the cross-domain experiment (see Figure 5) seems to be a special case of style transfer. In this sense, this paper should also include comparisons to the style-transfer methods.

References:
Ensemble of Exemplar-SVMs for Object Detection and Beyond, Malisiewicz et al., In ICCV 2011.

**Summary Of The Paper:**

This paper studies the challenging problem of transferring a pre-trained GAN from the source domain to a target domain with only one example available. To achieve the source-to-target domain adaptation while being able to synthesize diverse samples, this paper proposed a novel method called GenDA. The key idea is to freeze the generator and discriminator backbones while fine-tuning the linear layer added on top. Convincing experiment results have been demonstrated on the standard face and building benchmarks.

**Summary Of The Review:**

Overall, this is a very interesting paper with convincing results. I believe the key idea from this paper can have a broader impact on the community (e.g., 3D and video synthesis) in the future. I recommend accepting this paper but would kindly ask the authors to incorporate the suggested discussions in the final version.

---

> ### Author Response · Authors · 2021-11-17
> **Response to Reviewer NUUH**
>
> **Q1. Ablation study on different architectural designs of attribute adaptor and classifier.**
>
> Thanks. As suggested, we conduct ablation studies on the architecture of the attribute adaptor and the attribute classifier, respectively. The results are also included in Tab. A2 of *the revised version*. Specifically, we replace the lightweight adaptor and classifier with a two-layer MLP independently. Experimental results suggest that the synthesis performance becomes worse when increasing the capacity of either network due to the overfitting problem. This in turn verifies the effectiveness of our simple design.
>
> **Q2. Discussion with Exemplar SVM.**
>
> We have included a discussion in Sec. 2.2 of *the revised version*.
>
> **Q3. The limitation is not clearly stated.**
>
> Thanks for pointing this out. We have updated the limitation to *explicitly* state that our method would fail when the inter-subclass variation is huge in Sec. 4 of the revised paper.
>
> **Q4. Include the comparisons to the style-transfer methods.**
>
> We compare our GenDA with the recent style-transfer method, SWAG (Wang *et al.* CVPR21). Results are shown in Fig. A4 of *the revised version*. It can be seen that GenDA could transfer the style in a more harmonious way, while SWAG transfers the color inconsistently, leading to obvious artifacts.

---

### Official Review · Reviewer_Zh5e · 2021-11-01

**Correctness:** 1
**Technical Novelty And Significance:** 2
**Empirical Novelty And Significance:** 2
**Recommendation:** 3
**Confidence:** 5

**Main Review:**

The paper is generally well written and understandable. The paper conveys the main ideas of the work to the reader and I believe I got a good understanding from what the authors did by reading the paper. It seems very doable to reproduce the results on the paper alone.

The two technical main ideas are simple, but somewhat novel and can be considered a contribution to the field. Most, I like the idea of reusing a pre-trained discriminator. I am not aware of similar work and this seems to be a creative idea.

On the downside, I do not like the idea of the adaptor layer. The adaptor does not seem powerful enough to match the complexity of the task. There are multiple concerns here. The layer is too early in the network and seems to manipulate z-space rather than w-space (or some of the other available latent spaces later in the network). The layer is not powerful enough to do a more complex computation. Also, naturally, domain transfer generally means that the generator does not have the capabilities to generate images in the new domain. What is this network really doing by only manipulating a z-vector? If the z-vector is not split, it is hard to imagine the generator has the range to produce the desired results. If the z-vector is split, this could be a bit better, but also carries more risks. As it is, we can only observe a limited form of domain transfer.

One concern regarding the results is Figure 4. Visually all faces look really nice. But the task is not that clear and it's really difficult to see a relationship between input and output. This is not a result that should be considered domain adaptation. I would strongly suggest that all results transferring faces to other faces (but both input and output are in the same domain) should be removed from the paper. While this uses the same software system, it is essentially a different task that has been addressed better by many other papers. For example, attribute-based GAN editing papers can do that as well and there are multiple of them. I am pretty sure the authors do not want to go down that route.
The challenging domain transfer results, e.g. Figure 5, are indeed what I was hoping to see more of. Unfortunately, these results do not look particularly good. We can mainly observe a slight adaptation of all results shifting to a particular color scheme. It would also suggest to synchronize the examples. I think this is would be more insightful if we can see the same inputs transferred to different domains. I also feel the results are hard to interpret in multiple figures (e.g. 3), because I do not understand where the corresponding images in the original domain are. Maybe the input changes a lot for the challenging examples. I am afraid that this is a considerable omission, because the images in the input domain are such an essential component of the results. How can anybody judge these results without seeing the input images? In addition, there should be corresponding results shown. E.g., the last two rows in Figure 5 should correspond to each other. I do like the result in Figure 2. In a future version of the paper it would be good to see more of these interesting results.

The evaluation of this type of work is primarily visual. The metrics are not especially revealing if the results are good. It's good to have them, but I feel the authors still need quite a bit of work to make the system function as desired. My recommendation would be to revisit the adaptor design and find something better.


**Summary Of The Paper:**

The paper proposes to do one-shot domain adaption by fine tuning a state of the art network StyleGAN. The paper introduces a few techniques to do so. First, the paper proposes to add a lightweight attribute adaption layer in the beginning of StyleGAN to modify the latent code. An existing latent code can be modified to give a new latent code that is in the new domain. Second, the paper proposes a novel type of discriminator reuse. A discriminator has its features frozen, but a new classification head is added. The paper shows various visual results.

**Summary Of The Review:**

I am very positive about the potential of this work, but it needs a substantial improvement.

---

> ### Author Response · Authors · 2021-11-17
> **Response to Reviewer Zh5e Part-1**
>
> **The setting and challenge of few-shot generative domain adaptation.**
>
> Before addressing the concerns, we would like to reaffirm the setting of few-shot generative domain adaptation.
>
> It is a relatively new task, but has been formally defined and explored in the previous literature [Mo et al., 2020; Ojha et al., 2021]. Task of few-shot generative domain adaptation aims at “transferring a generative model that is well-trained on the source domain to a target domain with limited training data”, **rather than performing the image-to-image translation**. Namely, both before and after training, the model synthesizes images by only taking random noises as the inputs. Therefore, there is no concept of **input images** in this task.
>
> For a generative model, **the key evaluation criteria are the synthesis quality and diversity**. In other words, we don’t want the transferred model to simply memorize the training data, instead, the transferred model should capture the pattern of the adapted distribution and generate new samples. It is only a *byproduct* that our approach can maintain some semantics between the images synthesized from the same latent code before and after adaptation. We have provided sufficient qualitative and quantitative experimental demonstration on the effectiveness of our approach in Tab. 1, and Fig. 2.
>
> The main challenge of this task is that, under the few-shot setting, it is difficult to learn a new generative model for photo-realistic, and more importantly, highly-diverse synthesis. Especially, when the number of images in the target domain is reduced to one, this task becomes extremely challenging and all the previous approaches would fail (*i.e.*, the model after adaptation collapses to one mode). One of our main contributions is that **GenDA enables the task of generative domain adaptation under such a challenging setting**.
>
>
> **Q1. The adaptor does not seem powerful enough to match the complexity of the task.**
>
> We have conducted experiments on adapting the same source model to a wide range of target domains, where each domain only contains one reference image. As shown in Fig. 2 and Tab. 1, our approach is efficient and effective. We significantly surpass the second competitor from both the quality and the diversity perspectives.
>
> We also visualize the latent representations after adaptation in Fig. A3. Obviously, the original latent space is successfully transformed to a new space regarding the target sample. This verifies that the proposed adaptor is powerful enough.
>
> Meanwhile, due to the challenging setting of only having one image from the target domain, a more complex structure may aggravate the overfitting problem, which is shown in Tab. A2. Instead, our efficient design alleviates such a problem, resulting in a satisfying domain adaptation.
>
> **Q2. manipulate $\mathcal{Z}$-space rather than $\mathcal{W}$-space.**
>
> This is a misunderstanding. In all the experiments, we always use the $\mathcal{W}$-space of StyleGAN for domain adaptation. **Implementation** in Sec. 3 (on Page 4) of the submission already explains this. We denote the latent space as $\mathcal{Z}$ in Sec. 2, just following the general formulation of deep generative models.
>
> **Q3. The task of Figure 4 is not that clear and it’s really difficult to see the relationship between input and output.**
>
> First of all, there are no input images in our task. Same as other experiments, the task of Figure 4 is to adapt a source GAN model to a target domain with only one/two images (*i.e.*, the first column). The top row shows the images synthesized by the source model, which are used for reference *only*. All samples from the remaining rows are synthesized using the same latent codes as those in the first row. As already pointed out in the caption, (1) pairing “the woman with eyeglasses” with “a man with eyeglasses”, GenDA transfers the “eyeglasses” attribute and maintains the “gender” attribute from the source model, (2) pairing the same woman with “another woman without eyeglasses”, GenDA transfers the "female’’ attribute and maintains the "eyeglasses’’ attribute from the source model.
>
> **Q4. All results transferring faces to other faces should be removed from the paper.**
>
> Actually, human faces also consist of different domains, like “male” and “female” can be viewed as two different domains. We follow the standard setting in the few-shot generative domain adaptation task introduced in [Ojha et al., 2021], which is also included in the submission as a baseline.
>
> In addition, GenDA can achieve generative domain adaptation regarding both “attribute-level” (*e.g.*, child) and “style-level” (*e.g.*, painting style) semantics, as shown in Fig. 3. This is a completely different task to the attribute-based GAN editing.

---

> > ### Author Response · Authors · 2021-11-17
> > **Response to Reviewer Zh5e Part-2**
> >
> > **Q5. I think this would be more insightful if we can see the same inputs transferred to different domains in Figure 5.**
> >
> > Again, there are no input images in our task. But as suggested, we feed the same latent codes into the models before and after adaptation to see how the synthesized images change. This is included in Fig. A6. Here, the synthesis from the source domain is only shown as the reference. We can tell that our GenDA is able to transfer the most distinguishable attributes of the target domain, but almost maintains other semantics (*e.g.*, the face pose and the church shape).
> >
> > **Q6. I also feel the results are hard to interpret in multiple figures (e.g. 3), because I do not understand where the corresponding images in the original domain are.**
> >
> > Please refer to Q5, there are no input images in our task. The requested results are included in Fig. A5. Our task setting is clearly different from image-to-image translation or style transfer.
> >
> > **Q7. The evaluation of this type of work is primarily visual.**
> >
> > We follow the same evaluation metric in previous work [Ojha et al., 2021]. Here, FID measures the distributional distance between the synthesized data and the real data. This is widely used to evaluate generative models. Moreover, we also include the precision and recall metrics [Kynkaanniemiet al., 2019], which are primarily proposed to evaluate the synthesis diversity of a generative model. All metrics are well-defined and commonly used for GAN evaluation. All quantitative results demonstrate the superiority of our approach.

---

> > > ### Author Response · Authors · 2021-11-23
> > > **Any further comment?**
> > >
> > > Dear Reviewer Zh5e,
> > >
> > > Thank you so much for your effort in reviewing this submission! We have tried our best to address your concerns above the reply. Would you mind taking a look and letting us know what you think?
> > >
> > > Feel free to let us know if there is anything unclear or so. We are happy to clarify them.
> > >
> > > Best,
> > > Authors

---

> > > > ### Comment · Reviewer_Zh5e · 2021-11-27
> > > > **Still too many issues**
> > > >
> > > > The rebuttal does not really address the core concerns. I still do not understand why the technique should be called domain adaptation. The technique should be called domain collapse or domain restriction. The technique takes points inside the domain and maps them to a smaller subset that looks similar to the input. It's really just restricting the domain to a much smaller subset. In the end, the generated images are inside the domain of the generator. Domain collapse can be used for editing (as shown in the paper) but there are better methods. Domain collapse cannot be used for challenging tasks that other researchers call domain adaptation and that is evident in the results.

---

> > > > > ### Author Response · Authors · 2021-11-28
> > > > > **Response to further comments**
> > > > >
> > > > > Thanks for your comments.
> > > > >
> > > > > First, **the task studied in this work is well-defined in the literature of generative modeling and raises wide attention in the past year**. Please refer to [1] (CVPR'20) [2] (CVPRW'20) [3] (NeurIPS'20) [4] (CVPR'21), which are all published at top conferences recently. These references are already cited and discussed in our submission, and we just follow them about the task name. Thus, with all due respect, **we *disagree* that "why the technique should be called domain adaptation" can be the core concern**.
> > > > >
> > > > > Second, our task is far beyond domain collapse or domain restriction. For example, the sketches in the last row of Fig. 2 in our submission **cannot** be generated by the generator trained on the original domain. Under such a case, **"collapse" or "restriction" seems inaccurate**. Transferring a face model to synthesize only faces wearing eyeglasses (or baby faces) is only a sub-task (or say, byproduct) of our approach. From this point of view, with all due respect, **we *disagree* that "the technique should be called domain collapse or domain restriction"**.
> > > > >
> > > > > Third, about the concern "domain collapse can be used for editing (as shown in the paper) but there are better methods". To the best of our knowledge, existing approaches can **barely transfer a generative model to synthesize images regarding a second domain *referring to as few as one image* and achieve *high diversity***. Please correct us by naming a few *"better methods"*.
> > > > >
> > > > > Hope that the above discussions address your concerns.
> > > > >
> > > > > [1] MineGAN: Effective Knowledge Transfer from GANs to Target Domains with Few Images. Wang et al. CVPR'20.
> > > > >
> > > > > [2] Freeze the Discriminator: a Simple Baseline for Fine-Tuning GANs. Mo et al. CVPRW'20.
> > > > >
> > > > > [3] Few-shot Image Generation withElastic Weight Consolidation. Li et al. NeurIPS'20.
> > > > >
> > > > > [4] Few-shot Image Generation via Cross-domain Correspondence. Ojha et al. CVPR'21.

---

> > > > > > ### Comment · Reviewer_Zh5e · 2021-11-29
> > > > > > **I cannot follow the authors argumentre**
> > > > > >
> > > > > > I understood that the paper takes a w latent code and maps it to another one. I also understood that the generator network does not change.  So almost per definition, if the generator network does not change, the domain also does not change.
> > > > > >
> > > > > > The authors also seem to agree that adding sunglasses is not a main focus, but only a byproduct. Then why does so much of the paper and the evaluation focus on this byproduct? This was one of my original complaints.
> > > > > >
> > > > > > I also agree with the authors that there are reasonable results for this byproduct. But this is what I call domain restriction because you map all latent codes to only latent codes of people wearing sunglasses. All the results were already in the original latent space. However, there are GAN editing methods that can take multiple labeled images as input and compute a linear edit vector to add sunglasses. This could be done in a single shot method, e.g. by computing an editing vector from a mean face to the sunglass face, but that would be contrived. Even so, do you compare to this baseline? That is what others did before going many years back. But this is not even the main issue for me. There is no problem learning a sunglass edit or similar, and one can easily label 100s of images of people with sunglasses. It’s not difficult. So this is a contrived problem and not a single shot problem. Maybe someone is interested in this, possible. It’s fine for the authors to argue so. But I cannot understand why this part of the paper should be called domain adaption.
> > > > > >
> > > > > > There is another part of the paper that shows results for attempted true domain adaption, that is also correct. The authors argue that the results are good, but I argue that the results are poor and also not really evaluated. I do agree that the sketch results look fairly reasonable. Many of the other results are not good though. How could the generator have learned to produce images in other domains without any training? Just by mapping to regions in the latent space that the mapping network usually does not access? If the results would look good, then I would fully agree with the authors. Then they would have found a remarkably simple and surprising method for domain adaption.
> > > > > >
> > > > > > I mainly tried to be helpful to the authors. I provided a likely explanation as to why the results in challenging cases are not that good. Because the generator does not change, you don’t have the required degrees of freedom to perform domain adaption. You may  consider moving to an extended latent space (I do not recommend this) or changing the generator. I would also like to refer to a recent paper StyleAlign. While oversimplifying a bit, i feel the paper also gives results that seem to indicate that most of the transfer results come from changing the generator and not from changing the mapping network. This has nothing to do with the evaluation. I just try to convince the authors that changing the generator is likely necessary for domain adaption and that they should improve on their initial idea.

---

> > > > > > > ### Author Response · Authors · 2021-11-30
> > > > > > > **Thanks for the detailed comments**
> > > > > > >
> > > > > > > **Review:** "if the generator network does not change, the domain also does not change".
> > > > > > >
> > > > > > > **Answer:** This may be a misunderstanding. We have emphasized in previous response that **there are also multiple domains even for faces**. For example, real faces, sketches, cartoons all belong to different domains. Babies, children, adults, elderly also belong to different domains. Under such a background, it is possible to transfer the generator from one domain to another without touching the synthesis network. Here is the reason. The convolutional kernels of early layers are more likely to render basic shapes, like circle (low-level) and eyes (high-level). The convolutional kernels of later layers are more likely to render color. Prior arts agree to call these concepts as **prior knowledge**. The latent codes of StyleGAN tend to tell the synthesis network how to organize these basic concepts to produce a high-quality synthesis. **In this work, we propose to *reuse* these knowledge for domain adaptation**. We argue that, by **reorganizing** these concepts, we manage to transfer the domain. Hence, **even the generator network does not change, the domain *indeed* changes**, which has been demonstrated both qualitatively and quantitively. We **have also discussed our limitation in the submission**, which is that we cannot transfer domains that are highly different from each other, like faces to churches. That is because these is **only one** training image, **making it hard to learn new concepts for the synthesis network**.
> > > > > > >
> > > > > > > **Reviewer:** "But I cannot understand why this part of the paper should be called domain adaption."
> > > > > > >
> > > > > > > **Answer:** Because this is a **standard protocol (or say, experimental setting)** in the previous literature. Please refer to Ojha et al. (CVPR 2021), which is currently the state-of-the-art approach in this field. It is **not fair** to question a well-defined task (defined by prior arts) by just raising "I cannot understand why xxx is called xxx". By the way, with all due respect, it seems unfair to reject ResNet by saying "I cannot understand why the task should be called image classification and object detection".
> > > > > > >
> > > > > > > **Reviewer:** "How could the generator have learned to produce images in other domains without any training? Just by mapping to regions in the latent space that the mapping network usually does not access? If the results would look good, then I would fully agree with the authors. Then they would have found a remarkably simple and surprising method for domain adaption."
> > > > > > >
> > > > > > > **Answer:** We have explained many times on this "how". That is because **we manage to *reuse* the learned knowledge as much as possible**. Both the qualitative and quantitative results point to this conclusion. Also, the reviewer has agreed that ***If the results would look good, then I would fully agree with the authors. Then they would have found a remarkably simple and surprising method for domain adaption***. This is just what we want to claim in this paper. **The results are good and also appreciated by the other two reviewers**. Please be specific **on which part(s) the results are not good**? If you insist on judging a submission **just because of not believing**, then we have no further comments.
> > > > > > >
> > > > > > > **Reviewer:** "Because the generator does not change, you don’t have the required degrees of freedom to perform domain adaption." and "I just try to convince the authors that changing the generator is likely necessary for domain adaption and that they should improve on their initial idea."
> > > > > > >
> > > > > > > **Answer:** Please refer to the first question-answer. Under the challenging one-shot setting in domain adaptation, **reusing the prior knowledge** is critical. If changing the entire generator with the naive fine-tuning, we even fail to transfer faces to sketches, let alone transfer faces to churches. StyleAlign studies a different topic, where they do not really care about the number of training images. **We agree that when there are sufficient training samples, learning the entire generator may lead to a better performance.** **However, this conclusion is orthogonal to the goal of our work, where we provide an *elegant and effective* solution to the one-shot setting.** Everyone should agree that, sometimes, we may only have one reference image, like Mona Lisa.

---

> > > > > > > > ### Comment · Reviewer_Zh5e · 2021-11-30
> > > > > > > > **Result quality**
> > > > > > > >
> > > > > > > > To be more specific, all results where the target domain is given by a painting are not good. The method successfully takes on the colors of the target image, but fails to take on the distinct characteristics of a painting, such as brush strokes. The results look scary. One can simply  look at the results in the first figure for the Van Gogh example. Full screen and then look at it for 5 minutes. Does the result look like a painting? No. Look at the adaption to Dix.
> > > > > > > >
> > > > > > > > If the other reviewers indeed think the results are good I can accept that as a different opinion. But I do not think the domain adaption was successful in the important cases (paintings) and I am confident in my ability to judge visual quality. As I wrote before, I do appreciate the quality of the sketch transfer in Figure 2. This is a successfully and non-trivial case of domain adaption that is included in the paper. Also to amend that, the Superman example in figure 5 is reasonable considering the complexity of the task. But look at the top two rows of Figure 5 full screen. This is unreasonable.
> > > > > > > >
> > > > > > > > I know there is no chance to convince the authors and it’s fine that the authors want to insist in their opinion. At this point, I just want to make sure the authors understand my opinion. It’s up for the area chairs to decide, since the discussion is getting repetitive.

---

### Decision · Program_Chairs · 2022-01-20

**Decision:**

Reject

**Comment:**

This work was the subject of significant back and forth (between authors and reviewers, but also between reviewers & myself) due to the wide range of opinions. Two of the reviewers have found this work below the bar: they have provided multiple reasonings that I would rather not repeat here. The third reviewer found this work more compelling and argued for its acceptance. My attempts at reaching a consensus have yielding the following conclusions:

  * There's agreement that one-shot generation is indeed a challenging task
  * Some of the results are indeed impressive, but many results are not compelling.
  * The rebuttal addressed some of the concerns (e.g. visualization of latents), but some issues are unaddressed (e.g. more motivation, explanation of why the proposed method works better)
  * One of the reviewers has argued rather forcefully that the work doesn't quite do domain adaptation in the typically understood sense. Moving beyond definitions of domain adaptation, the same reviewer was not very convinced by the quality of the results themselves.
  * The reviewer most positive about this work agrees that this work only explores a limited form of domain transfer. They argued that some of the potential applications of this work do make the submission interesting.

Fundamentally, the discussion did not necessarily resolve the differences in opinion one way or another. Ultimately, all 3 reviewers believe that it would fine if this work was not accepted to ICLR at this time, despite some of the interesting results and promise. Given the discussion and this mildest consensus, I am inclined to recommend rejection too. I do think there's a substantial amount of constructive feedback in the reviews that would make a subsequent revision of this work quite a bit better.